# Efficient Model-Based Reinforcement Learning Through Optimistic Thompson Sampling

**Jasmine Bayrooti**
University of Cambridge
jgb52@cam.ac.uk

**Carl Henrik Ek**
University of Cambridge
Karolinska Institutet
che29@cam.ac.uk

**Amanda Prorok**
University of Cambridge
asp45@cam.ac.uk

## Abstract

Learning complex robot behavior through interactions with the environment necessitates principled exploration. Effective strategies should prioritize exploring regions of the state-action space that maximize rewards, with optimistic exploration emerging as a promising direction aligned with this idea and enabling sample-efficient reinforcement learning. However, existing methods overlook a crucial aspect: the need for optimism to be informed by a belief connecting the reward and state. To address this, we propose a practical, theoretically grounded approach to optimistic exploration based on Thompson sampling. Our approach is the first that allows for reasoning about *joint* uncertainty over transitions and rewards for optimistic exploration. We apply our method on a set of MuJoCo and VMAS continuous control tasks. Our experiments demonstrate that optimistic exploration significantly accelerates learning in environments with sparse rewards, action penalties, and difficult-to-explore regions. Furthermore, we provide insights into when optimism is beneficial and emphasize the critical role of model uncertainty in guiding exploration.

## 1 Introduction

Reinforcement Learning (RL) is recognized as a promising approach to solving complex sequential decision-making tasks in robotics (Levine et al., 2016). However, many popular RL algorithms require millions of interactions with the real world to train effective policies (Schulman et al., 2017; Mnih et al., 2015). This sample inefficiency is primarily due to the central challenge of balancing exploration, gathering information about the world, with exploitation, maximizing rewards given current knowledge (Sutton & Barto, 2018; Szepesvári, 2022). Sample inefficiency is especially prohibitive for RL applied to robotics due to the high cost and potential wear on physical systems.

Efficient RL requires a careful balance between exploration and exploitation, and some approaches are better suited for this than others. Our work centers on model-based RL, a class of algorithms where agents build a predictive model of the environment dynamics from which they derive a policy. Many works have demonstrated impressive sample efficiency using model-based RL by leveraging the learned dynamics model to simulate future outcomes, thereby allowing agents to plan strategically and optimize the controller using fewer real-world trials (Ibarz et al., 2021; Chua et al., 2018; Janner et al., 2019; Yang et al., 2020; Sekar et al., 2020; Hansen et al., 2024). However, learning a model of the environment is a double-edged sword: a faithful model enhances efficiency by reducing the number of interactions needed in the real world while a biased model severely hampers performance by misguiding the policy (Gu et al., 2016). Planning with *uncertainty-aware* dynamics reduces the effects of model errors (Deisenroth & Rasmussen, 2011; Deisenroth et al., 2013; Gal et al., 2016) and allows model-based RL algorithms to reach state-of-the-art asymptotic performance on benchmark control tasks (Chua et al., 2018; Janner et al., 2019). Furthermore, by learning a probabilistic dynamics model, model-based RL enables directing exploration in a principled manner towards unseen parts of the state-action space.

In this work, we focus on *optimistic exploration* using model-based RL with an uncertainty-aware model. In a subtle departure from classical exploration, which prioritizes transitions with high uncertainty under the model, we define optimistic exploration as favoring transitions likely to yield high

rewards. This approach aligns well with the RL objective of maximizing cumulative rewards (Sutton & Barto, 2018). Notable works unite optimistic exploration with uncertainty-aware model-based RL to hallucinate plausible, optimistic training experiences for superior robustness and sample efficiency (Curi et al., 2020; Sessa et al., 2022), however these formulations ignore joint uncertainty between the reward and dynamics. They also assume knowledge of a reward function, which can be impractical in real-world applications due to the difficulty of specifying dynamic, variable, and subjective reward functions accurately a priori. While learning the reward function is a straightforward remedy, there is a dearth of efficient and principled approaches to integrating reward learning into optimistic exploration for model-based RL.

**Contributions.** Our primary contribution is the first practical model-based RL algorithm, called Hallucination-based Optimistic Thompson sampling with Gaussian Processes (HOT-GP), that can be used with state-of-the-art off-policy RL algorithms for *principled* optimistic exploration. Inspired by the premise that efficient exploration should be informed by joint uncertainty over state and reward distributions, we propose a GP model to maintain this joint belief in order to simulate transitions that are simultaneously plausible under the learned dynamics and optimistic with respect to the estimated reward. We evaluate this approach on a set of MuJoCo (Todorov et al., 2012) and VMAS (Bettini et al., 2022) continuous control tasks. The results reveal that HOT-GP matches or improves sample efficiency on standard benchmark tasks and substantially accelerates learning in challenging settings involving sparse rewards, action penalties, and difficult-to-explore regions. As these findings validate our hypothesis, our secondary contribution is an empirically-supported argument for maintaining a belief over *both* the reward and state distributions, as this is key for effective optimistic exploration. Moreover, we study the factors that influence the utility of optimism and underscore the important role of model uncertainties in shaping exploration.

## 2 RELATED WORK

**Model-based RL** methods are promising for complex real-world decision problems due to their potential for data efficiency (Kaelbling et al., 1996). Selecting the appropriate model is critical, as it must facilitate effective learning in both low-data regimes (during early stages of training) and high-data regimes (later in training). This requirement naturally aligns with the advantages of Bayesian models. Non-parametric models like Gaussian processes (GPs) provide excellent performance in settings with limited, low-dimensional data (Deisenroth et al., 2013; Deisenroth & Rasmussen, 2011; Kamthe & Deisenroth, 2018). Neural network predictive models have also been effectively used for tasks with non-smooth dynamics (Nagabandi et al., 2018) and high dimensions like the Atari suite (Oh et al., 2015; Kaiser et al., 2020). As deterministic neural networks tend to overfit to data in the early stages of training, uncertainty-aware neural network models have been shown to achieve better performance with algorithms such as PETS (Chua et al., 2018) and MBPO (Janner et al., 2019). However, designing scalable Bayesian neural networks remains an open problem (Roy et al., 2024; Guo et al., 2017; Osband, 2016) and popular approximations using dropout (Gal et al., 2017) and ensembles (Lakshminarayanan et al., 2017; Chua et al., 2018) fall short of the precise uncertainty quantification that GPs provide. In order to benefit from both the high-capacity function approximation of neural networks and probabilistic inference of GPs, in this work we model the joint dynamics and reward functions using a GP with a neural network mean function (Iwata & Ghahramani, 2017).

**Thompson sampling** is a provably efficient exploration algorithm in RL (Thompson, 1933). This approach implicitly balances exploration and exploitation by sampling statistically plausible outcomes from the posterior distribution over dynamics models and selecting actions based on the sampled outcomes (Russo & Van Roy, 2014; Russo et al., 2018). Thompson sampling has been studied on tabular MDPs (Osband et al., 2013) and extended in theory to continuous state-action spaces with sublinear regret under certain conditions (Chowdhury & Gopalan, 2019).

**Optimism in the face of uncertainty** is another theoretically grounded approach to guide exploration that selects actions that optimize for optimistic outcomes. The motivation is to promote exploration guided by beliefs about future value to facilitate efficient learning. While optimism has been extensively studied in tabular MDPs (Brafman & Tennenholtz, 2002; Jaksch et al., 2010) and linear systems (Jin et al., 2020; Abbasi-Yadkori & Szepesvári, 2011; Neu & Pike-Burke, 2020), optimism in continuous state-action MDPs with non-linear dynamics remains less explored. GP-

UCRL (Chowdhury & Gopalan, 2019) is one such theoretical approach that targets dynamics models that lie in a Reproducing Kernel Hilbert Space. Another is LOVE (Seyde et al., 2021), which uses an ensemble of networks to direct exploration towards regions with high predicted potential for long-term improvement. TIP (Mehta et al., 2022) also employs optimism by selecting actions according to an optimistic acquisition function. Curi et al. (2020) proposes a reparameterization of the dynamics function space in order to reduce optimistic exploration to greedy exploitation with the H-UCRL algorithm. H-UCRL has sublinear regret bounds under certain conditions and demonstrates efficient exploration on deep RL benchmark tasks with action penalties. The H-MARL algorithm extends this to the multi-agent setting for general-sum Markov games and introduces a practical approximation for generating optimistic states (Sessa et al., 2022). While these methods all derive useful optimistic exploration strategies for RL, they ignore the relationship between uncertainty around dynamics and uncertainty around the reward. Moreover, TIP, H-UCRL, H-MARL use the ground-truth reward function throughout training. Our approach, inspired by H-UCRL (Curi et al., 2020), improves optimistic hallucination for model-based RL by learning and leveraging the joint uncertainty over state transitions and associated rewards.

**Reward learning** is commonly applied in RL for reward shaping (Hu et al., 2020). This is closely related to exploration, as intrinsic rewards learned through curiosity-driven methods help guide exploration by reducing uncertainty over the agent's knowledge of the environment (Pathak et al., 2017). Reward learning also plays an important role in model-based RL. Receding-horizon planning approaches like PETS (Chua et al., 2018), Dreamer (Hafner et al., 2020; 2019; 2023), and TD-MPC (Hansen et al., 2022; 2024) use estimated rewards to evaluate and optimize sequences of actions, while model-based policy learning approaches like MBPO (Janner et al., 2019) use estimated rewards to inform policy updates. Some of these approaches explicitly account for uncertainty in the learned reward function (Chua et al., 2018; Janner et al., 2019; Chowdhury & Gopalan, 2019), however they treat it as independent from the uncertainty in the dynamics. We argue that learning an uncertainty-aware reward-dynamics function and strategically leveraging the joint uncertainty leads to the most principled and effective optimistic exploration.

## 3 PROBLEM STATEMENT

We consider a stochastic environment with states $s \in \mathcal{S} \subseteq \mathbb{R}^p$ and actions $a \in \mathcal{A} \subseteq \mathbb{R}^q$. Given the current state $s_t$ and action $a_t$, the agent transitions to the next state $s_{t+1}$ and incurs reward $r_t$ with

$$\begin{pmatrix} s_{t+1} \\ r_t \end{pmatrix} = f(s_t, a_t) = \begin{pmatrix} f_t(s_t, a_t) \\ f_r(s_t, a_t) \end{pmatrix} \tag{1}$$

for transition dynamics $f_t : \mathcal{S} \times \mathcal{A} \to \mathcal{S}$ and reward function $f_r : \mathcal{S} \times \mathcal{A} \to \mathbb{R}$. Our objective is to learn optimal control for this system within episodes of finite time horizon $T$. To control the agent, we learn a deterministic or stochastic policy $\pi$ that selects actions given the current state. For ease of notation, we write the policy deterministically as $\pi : \mathcal{S} \to \mathcal{A}$ so that $a_t = \pi(s_t)$. We consider a specific transition dynamic $\hat{f}_t$ and reward function $\hat{f}_r$, denoted with a hat to indicate realizations that are drawn from random functions. The performance of the policy is measured by the sum of accumulated rewards during an episode,

$$J(\pi, \hat{f}_t, \hat{f}_r) = \left[ \sum_{t=0}^{T} \hat{f}_r(s_t, a_t) \right], \quad \text{s.t.} \quad a_t = \pi(s_t), s_{t+1} = \hat{f}_t(s_t, a_t). \tag{2}$$

The optimal policy $\pi^*$ maximizes performance over the true dynamics and reward function with,

$$\pi^* = \arg\max_{\pi \in \Pi} J(\pi, f_t, f_r). \tag{3}$$

In model-based RL, the true dynamic $f_t$ is unknown and must be learned by interactions with the environment. In our setting, the ground-truth reward function $f_r$ is also unknown.

## 4 BACKGROUND

In this section, we introduce relevant background on model-based reinforcement learning and existing exploration strategies.

## 4.1 MODEL-BASED REINFORCEMENT LEARNING

When systems have unknown dynamics, we consider the dynamics $f_t$ to initially be random. Therefore, learning a model of the environment dynamics entails fitting an approximation $p(f_t)$ over the space of dynamics functions, given observations from the real system. The same approach can be used to estimate the true reward function $f_r$. We consider model-based RL algorithms that, on each iteration of training, roll out the current policy $\pi$ on the real system for an episode and update $p(f)$ using the newly observed transition data. Standard approaches use the model to simulate sequential transitions and use this data to update the policy (Sutton, 1990). We opt to update the policy based on short $k$-length model-generated rollouts branched from the state distribution of a previous policy under the true environment, as recommended by Janner et al. (2019) to reduce compounding model errors over extended rollouts. Algorithm 1 gives an overview of the process.

---

**Algorithm 1** Model-based Policy Optimization

---

1: **Require:** max environment steps $N$, model rollouts $M$, steps per rollout $T$, steps per model rollout $K$, initial state distribution $d(s_0)$, policy-learning algorithm `PolicySearch`
2: **Initialize:** policy $\pi$, reward-dynamics model $p(f)$, environment dataset $\mathcal{D}_{\text{env}}$
3: **while** $|\mathcal{D}_{\text{env}}| < N$ **do**
4:     /*Simulate Data*/
5:     Initialize model dataset $\mathcal{D}_{\text{model}} = \emptyset$
6:     **for** $m = 1, 2, \ldots, M$ **do**
7:         Sample $\hat{s}_0$ uniformly from $\mathcal{D}_{\text{env}}$
8:         **for** $k = 1, \ldots, K$ **do**
9:             Compute action $\hat{a}_k$ from $\pi(\hat{s}_k)$
10:            Select next state $\hat{s}_{k+1}$ and reward $\hat{r}_k$ using $p(f \mid \mathcal{D}_{\text{env}})$   ▷ This is algorithm-specific
11:            Append transition to buffer $(\hat{s}_k, \hat{a}_k, \hat{s}_{k+1}, \hat{r}_k) \to \mathcal{D}_{\text{model}}$
12:     /*Optimize Policy*/
13:     $\pi \leftarrow$ `PolicySearch`$(\pi, \mathcal{D}_{\text{model}})$
14:     /*Optimize Dynamics Model*/
15:     Start from initial state $s_0 \sim d(s_0)$
16:     **for** $t = 1, 2, \ldots, T$ **do**
17:         Compute action $a_t$ from $\pi(s_t)$
18:         Observe next state and reward $s_{t+1}, r_t = f(s_t, a_t)$
19:         Append transition to buffer $(s_t, a_t, s_{t+1}, r_t) \to \mathcal{D}_{\text{env}}$
20:     Retrain model $p(f)$ using $\mathcal{D}_{\text{env}}$

---

**Model Design.** We use an uncertainty-aware model to represent the prior distribution of dynamics models $p(f_t)$ aligned with the 1-step transition data in $\mathcal{D}_{\text{env}}$. Probabilistic models allow for exploiting correlations in observed data by generalizing to unseen states and actions. Popular approaches include probabilistic ensembles of neural networks, which provide mean $\mu(s, a)$ and confidence estimates $\Sigma(s, a)$ by averaging outputs over multiple trained models (Lakshminarayanan et al., 2017; Chua et al., 2018). Bayesian inference with neural networks is computationally expensive and it is an open question if parameter uncertainties lead to relevant posterior function estimates (Sharma et al., 2023; Roy et al., 2024). Alternatively, GPs provide distributions directly over the function space, enabling direct estimation of the posterior distribution $p(f_t \mid \mathcal{D}_{\text{env}})$ over dynamics models. Since GPs incorporate strong and interpretable prior beliefs, they naturally capture uncertainty due to limited data (epistemic uncertainty) and variability in the data (aleatoric uncertainty) (Rasmussen & Williams, 2006), which contributes to GPs' effectiveness in low-data regimes. GP priors, defined by a mean and covariance function, describe how the function varies around the mean. With flexible mean functions, GPs offer principled uncertainty estimates for complex functions (Iwata & Ghahramani, 2017; Fortuin et al., 2019; Fortuin, 2022).

## 4.2 EXPLORATION STRATEGIES

It is challenging to learn the reward-dynamics model while solving for optimal control. Performance depends on the utility of model-generated data for updating the policy and likewise the value of real experiences gathered by the policy for updating the model. Solving this problem efficiently requires balancing exploitation of current knowledge with exploration of the state-action space.

The mechanism for selecting the next model-generated state plays an important role, with different strategies incorporating varying degrees of uncertainty to encourage exploration. In this section, we provide an overview of the relevant exploration strategies, as categorized in Curi et al. (2020).

**Greedy Exploitation.** Most model-based RL approaches approximate $p(f_t \mid \mathcal{D}_{\text{env}})$ and maximize the policy greedily over the uncertainty in the dynamics model. They use this greedy policy:

$$\pi^{\text{Greedy}} = \arg\max_{\pi \in \Pi} \mathbb{E}_{p(f_t \mid \mathcal{D}_{\text{env}})} J(\pi, \hat{f}_t, \hat{f}_r). \tag{4}$$

For instance, GP-MPC (Kamthe & Deisenroth, 2018) and PETS (Chua et al., 2018) are MPC-based methods that use greedy exploitation and represent $p(f_t \mid \mathcal{D}_{\text{env}})$ with a GP and ensemble of neural networks respectively. Similarly, PILCO (Deisenroth & Rasmussen, 2011) uses a GP with moment matching to estimate $p(f_t \mid \mathcal{D}_{\text{env}})$ and greedy exploitation to optimize the policy. While such algorithms can converge to optimal policies under certain reward and dynamics structures (Mania et al., 2019), greedy exploitation is generally inefficient for exploration.

**Thompson Sampling.** This is a theoretically-grounded approach that offers a principled balance between exploration and exploitation (Russo et al., 2018; Chapelle & Li, 2011). Under Thompson sampling (Thompson, 1933), the policy is optimized with respect to a single model sample $\hat{f}$ on each episode,

$$\hat{f} \sim p(f \mid \mathcal{D}_{\text{env}}), \quad \pi^{\text{TS}} = \arg\max_{\pi \in \Pi} J(\pi, \hat{f}_t, \hat{f}_r). \tag{5}$$

This optimization problem is equivalent to greedy exploitation (equation 4) after sampling $\hat{f} \sim p(f \mid \mathcal{D}_{\text{env}})$. Under certain assumptions, Thompson sampling satisfies strong Bayesian regret bounds (Osband & Van Roy, 2016; 2017), theoretically positioning it as the optimal strategy.

**Hallucinated Upper-Confidence Reinforcement Learning (H-UCRL).** This is a tractable version of the UCRL algorithm (Jaksch et al., 2010), which optimizes an optimistic policy over the set of statistically plausible dynamics models. Curi et al. (2020) propose reparameterizing plausible dynamics models in order to optimize over a smaller class of variables. Specifically, they write dynamics functions as $\hat{f}_t(s, a) = \mu(s, a) + \beta \, \Sigma(s, a) \eta(s)$ for some function $\eta : \mathbb{R}^p \to [-1, 1]^p$ and optimize over $\eta(\cdot)$ instead. As this problem is still challenging to optimize, Sessa et al. (2022) propose sampling $n^j \sim \text{Uniform}([-1, 1]^p)$ for $j = 1, \ldots, Z$ and updating the policy with

$$\pi^{\text{H-UCRL}} = \arg\max_{\pi \in \Pi} \max_{\eta^j, j \in 1, \ldots, Z} J(\pi, \hat{f}_t, \hat{f}_r), \text{ s.t. } \hat{f}_t(s, a) = \mu(s, a) + \beta \, \Sigma(s, a) \eta^j. \tag{6}$$

Unlike greedy exploitation, this approach effectively optimizes an optimistic policy over the modified dynamics $\hat{f}_t$ in equation 6, although the reward function is assumed to be known. H-UCRL also does not model or utilize the joint uncertainty over transitions and rewards.

## 5 THE HOT-GP ALGORITHM

We introduce a new exploration strategy based on the insight that principled optimism should reason about uncertainty over transitions and rewards jointly. Importantly, this requires a joint model of the transition dynamics and reward function so that optimism about the reward can be connected to a distribution over states. Given such a model, there are many possible acquisition strategies. A natural approach uses Thompson sampling, which we adapt for optimistic exploration with our method, Hallucination-based Optimistic Thompson sampling with Gaussian Processes (HOT-GP).

### 5.1 MODEL STRUCTURE

Optimistic exploration implies biased exploration towards regions of the state-action space believed to yield high rewards. Thus, optimistic learning requires a reward-dynamics model that describes the correlation of state and reward. In this work, we use a Gaussian process (GP) prior to describe our belief over the product space of reward and state,

$$\hat{f} \sim \mathcal{GP}\left( \begin{pmatrix} \mu_s(\cdot) \\ \mu_r(\cdot) \end{pmatrix}, \begin{pmatrix} K_{ss}(\cdot, \cdot) & K_{sr}(\cdot, \cdot) \\ K_{sr}^{\text{T}}(\cdot, \cdot) & K_{rr}(\cdot, \cdot) \end{pmatrix} \right), \tag{7}$$

where $\mu_s : \mathcal{S} \times \mathcal{A} \to \mathcal{S}$ and $\mu_r : \mathcal{S} \times \mathcal{A} \to \mathbb{R}$ are the dynamics and reward mean functions and $K_{ss}, K_{sr}, K_{rr}$ represent covariance functions $(\mathcal{S} \times \mathcal{A}) \times (\mathcal{S} \times \mathcal{A}) \to \mathbb{R}$. Traditional GPs describe outputs as conditionally independent given the input and hence factorize across the state dimensions and the reward. However, we hypothesize that this assumption is unfounded in most RL tasks and furthermore antithetical to principled optimistic exploration. Therefore, we want to explicitly model the correlation structure between the dimensions of the predicted state, and more importantly, between the predicted state and its associated reward. While this correlation can naturally be included in the GP by parameterizing the full covariance structure across examples and output dimensions, this is computationally intractable due to cubic scaling with the size of the covariance matrix (Rasmussen & Williams, 2006). Instead, we adopt a linear model of coregionalization (Grzebyk & Wackernagel, 1994; Wackernagel, 2003) where the covariance between output dimensions is restricted to be linear combinations of the input covariance. This induces a Kronecker structured covariance function that allows for efficient factorization while still meaningfully directing optimistic exploration.

In most GP regression use cases, the mean function is a constant function. While this is a valid modeling assumption for simple data, the dynamics and reward structures we encounter in RL tasks vary in a non-constant manner across different regions of the input space. To better predict these complex transition-reward relationships, we use a GP model with a multi-layer perceptron (MLP) mean function, as in Iwata & Ghahramani (2017); Fortuin et al. (2019), and use the covariance function to learn the uncertainty, representing the residual around the MLP. This flexible model can capture intricate transitions while providing useful uncertainty estimates that can be leveraged in an optimistic fashion. The mean and covariance functions are learned jointly by performing approximate inference through optimizing a variational lower bound on the marginal likelihood, as done by Titsias (2009). This updates the posterior distribution $p(\hat{f} \mid \mathcal{D}_{\text{env}})$, which we use to obtain the posterior predictive distribution $p(s_{t+1}, r_t \mid s_t, a_t)$. Given the current state $s_t$ and action $a_t$, this multivariate Gaussian distribution reflects the model's predictions and uncertainty over the next state and reward.

## 5.2 USING THOMPSON SAMPLING IN THE GP ACQUISITION FUNCTION

Thompson sampling proscribes optimizing the policy using a single reward-dynamics model from the distribution of predictive models. A naive approach would sample from $p(s_{t+1}, r_t \mid s_t, a_t)$ until discovering a good model as a function of the predicted reward $r_t$. Instead, we observe that we can condition the reward value $r_t$ to be high, specifically greater than some minimum percentile threshold $r_{\text{min}}$ over the reward distribution, and sample once from this distribution. Crucially, this induces optimism into the sampling procedure because we only consider transitions where the next state is plausible under the dynamics model and whose associated reward we believe to be large.

To do this, we sample independent realizations from one non-parametric reward-dynamics model for each episode of length $T$, denoted as

$$\hat{f}^{(t:0 \to T)} \sim \prod_{t=0}^{T} p(s_{t+1}, r_t \mid s_t, a_t, r_t > r_{\text{min}}) \tag{8}$$

for $s_t$ sampled uniformly from $\mathcal{D}_{\text{env}}$ and $a_t = \pi(s_t)$ under the current policy. Then the optimistic Thompson sampling algorithm is

$$\hat{f}^{(0 \to T)} \sim p(s_{t+1}, r_t \mid s_t, a_t, r_t > r_{\text{min}}), \quad \pi^{\text{HOT-GP}} = \arg\max_{\pi \in \Pi} J(\pi, \hat{f}_t^{(0 \to T)}, \hat{f}_r^{(0 \to T)}). \tag{9}$$

Performing Thompson sampling following the joint distribution in equation 9 predicts both the next state and its associated reward in an optimistic manner. We argue that this would lead to a suboptimal exploration strategy as we are only interested in optimism over the state to the extent that it relates to optimism over the reward. To address this, we instead sample reward-dynamics model realizations using a two-step approach where only the reward is explored in an optimistic manner:

1. Sample an optimistic reward $\hat{r}_t$ from $p(r_t \mid s_t, a_t, r_t > r_{\text{min}})$, a truncated normal distribution, using inverse transform sampling or another sampling technique.
2. Compute the expected transition $\hat{s}_{t+1} = \mathbb{E}_{p(s_{t+1} \mid s_t, a_t, r_t = \hat{r}_t)}[s_{t+1}]$ conditioned on the sampled optimistic reward $\hat{r}_t$.

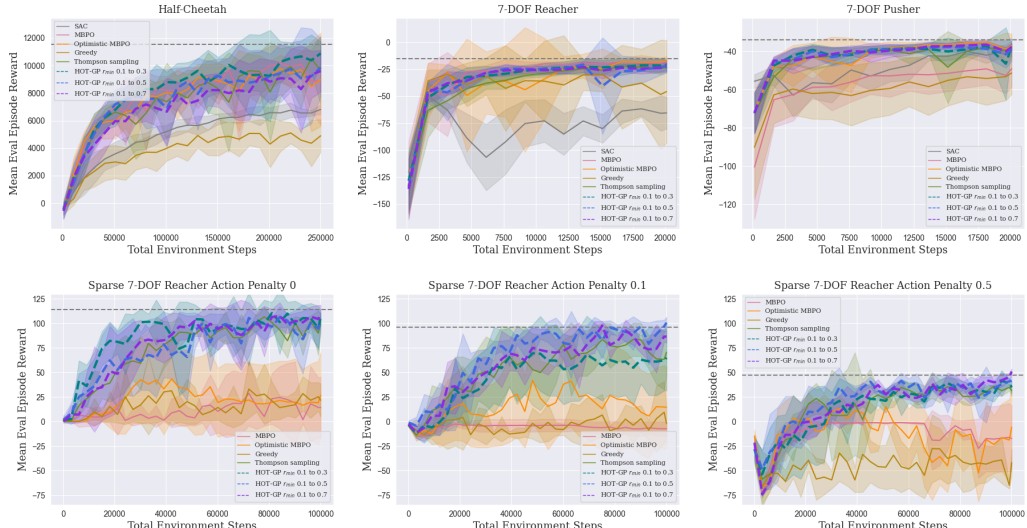

Figure 1: Learning curves for all MuJoCo tasks were averaged over 10 seeds except for the Sparse Reacher task, which used 5 seeds. HOT-GP demonstrates equivalent or superior sample efficiency and performance for all tasks considered. The dashed line denotes SAC performance at convergence within 1,000,000 environment steps (3,000,000 for Half-Cheetah).

This approach deviates from traditional Thompson sampling, and hence allows us to disentangle optimism over the state from optimism over the reward so that we can concentrate solely on optimism over the reward from the states. This aligns well with our goal of sample-efficient reinforcement learning as the objective is to maximize the reward in as few environment steps as possible.

## 6 EXPERIMENTS

We evaluate the performance of our proposed algorithm, HOT-GP, on continuous state-action control tasks by measuring the average sum of rewards accumulated during evaluation episodes. Since inaccurate reward estimates are likely initially, we increase $r_{\min}$ linearly throughout training to become more optimistic as the model becomes more reliable. In our experiments, we consider three variations of HOT-GP where $r_{\min}$ begins at 0.1 and increases to either 0.3, 0.5, or 0.7. We also consider the alternative Thompson sampling and greedy exploration strategies. Our Thompson sampling implementation follows the HOT-GP procedure but draws samples from the reward-conditioned distribution $p(s_{t+1} \mid s_t, a_t, r_t = \hat{r}_t)$ rather than taking the expectation. For greedy exploitation, we predict the next state and reward as the mean from the model.

First, we compare these approaches on standard MuJoCo benchmark tasks (Todorov et al., 2012) and extended MuJoCo sparse maze tasks from the D4RL suite (Fu et al., 2020) against additional model-free and model-based methods. Then we evaluate these approaches on a practical robotics task using the VMAS simulator (Bettini et al., 2022) and provide an analysis of the individual components. We provide algorithmic descriptions of HOT-GP and the model-based comparison methods in Appendix A. Details about training are discussed in Appendix B, specific tasks are described in Appendix C, additional evaluations are given in Appendix D, and code for reproducing the experiments can be found at https://github.com/jbayrooti/hot_gp.

### 6.1 MUJOCO BENCHMARKS

**Comparison Methods.** In these experiments, we consider HOT-GP with varying levels of optimism using SAC (Haarnoja et al., 2018) as the underlying `PolicySearch` algorithm, as done in Janner et al. (2019) with MBPO. We compare performance against SAC, Thompson sampling, greedy exploitation, and MBPO. SAC is a model-free actor-critic algorithm that is known to achieve better sample efficiency than DDPG (Lillicrap et al., 2015) on MuJoCo tasks (Chua et al., 2018). MBPO is

a model-based approach that uses an ensemble of probabilistic neural networks to model uncertainty as done in Chua et al. (2018). Predictions are drawn from a Gaussian distribution with diagonal covariance over the reward and dynamics: $p(s_{t+1}, r_t \mid s_t, a_t) = \mathcal{N}(\mu(s_t, a_t), \Sigma(s_t, a_t))$, where $\mu(s_t, a_t)$ and $\Sigma(s_t, a_t)$ are bootstrapped from the ensemble. For an optimistic baseline, we introduce an optimistic version of MBPO where the reward is sampled from $p(r_t \mid s_t, a_t, r_t > r_{\min})$ and expected next state $\mathbb{E}_{p(s_{t+1}|s_t, a_t)}[s_{t+1}]$ is selected. This is the same procedure as HOT-GP, however, due to MBPO lacking a joint model over the state and reward distributions, this implies a naive optimistic exploration as there is no explicit relationship between the predicted next state and reward.

**Standard Benchmarks.** The learning curves for all methods on Half-Cheetah, Reacher, and Pusher are given in the top row of Figure 1. In each case, the HOT-GP variants attain SAC-best performance with comparable or superior sample efficiency to other approaches. For instance, on Half-Cheetah, HOT-GP with $r_{\min}$ from 0.1 to 0.3 achieves the same final performance as SAC in 250,000 environment steps rather than 3 million. We observe that the deterministic approach used in greedy exploitation leads to slower learning on Half-Cheetah compared to uncertainty-aware methods, while showing no impact on performance on Pusher. This indicates that model uncertainty is key for efficient learning on Half-Cheetah, likely because the dynamics are more challenging to model, which makes model biases more detrimental to learning (Chua et al., 2018). Interestingly, uncertainty-aware learning alone is insufficient on Pusher as optimistic exploration methods enhance sample efficiency compared to MBPO and greedy exploitation. This is likely due to the task dynamics as the agent must navigate a landscape of negative rewards and sparse feedback. Optimistic exploration allows the agent to discover valuable actions more effectively by pursuing riskier options with potential to help reach the goal state. Next, we examine the impact of optimistic exploration in sparse reward settings more closely.

**Sparse Reacher.** The Sparse Reacher task provides reward $r_{\text{dist}}(s)$ that is nonzero only when the agent is within a small threshold of the target location, thus demanding clever exploration strategies to reach the target as there is no dense feedback. As in Curi et al. (2020), we introduce an action penalty of the form $r(s, a) = r_{\text{dist}}(s) - \rho \cdot \text{cost}(a)$ with the aim of penalizing random and encouraging local exploration. This is described in greater detail in Appendix C. We evaluate HOT-GP and the other comparison methods on Sparse Reacher using four different action penalty weights: $\rho = 0$, $\rho = 0.1$, $\rho = 0.3$, and $\rho = 0.5$, and present the results in the bottom row of Figure 1 as well as Figure 6 in Appendix D. HOT-GP demonstrates significantly superior sample efficiency over MBPO and optimistic MBPO in each case. Also, the asymptotic performance of HOT-GP matches the SAC-best performance for each action penalty while achieving it in a fraction of the environment steps. When $\rho = 0$, all HOT-GP variants perform robust optimistic exploration and successfully uncover the signal despite sparse rewards. When using the action penalty $\rho = 0.1$, HOT-GP with $r_{\min}$ from 0.1 to 0.3 performs marginally more poorly than with higher levels of optimism, suggesting that high levels of optimism may be required to overcome the prior against taking large actions and explore the reward signal $r_{\text{dist}}$. HOT-GP does this by relating the action-based penalty to the state to the overall reward through the joint model. The less efficient learning of optimistic MBPO highlights the critical importance of having a joint model of the reward and state distributions to facilitate efficient exploration. For $\rho = 0.3$ and $\rho = 0.5$, we observe that the cost term accounts for up to half of the overall reward. As a result, the HOT-GP variants explore much of the cost signal rather than $r_{\text{dist}}$, leading to similar performance in these settings.

**Sparse Maze.** In the Sparse Maze tasks, an agent must navigate through a maze to a randomized target location, receiving a reward of 1 when within a small threshold of the target location and zero in other cases. Similar to the Sparse Reacher task, efficiently solving such problems requires an effective exploration strategy due to the lack of dense feedback. We consider the simple U Maze and challenging Medium Maze environments, which are described in Appendix C. The results of HOT-GP and the comparison methods on these mazes are given in Figure 2. On the U Maze, HOT-GP attains equivalent sample efficiency and performance with the greedy and Thompson sampling strategies, while MBPO and optimistic MBPO are less stable. Greedy exploitation likely does well on this task because of the simple dynamics and reduced need for exploration. On the other hand, the Medium Maze presents a significantly more difficult exploration problem where greedy exploitation is insufficient to learn useful behavior. On the Medium Maze, HOT-GP with $r_{\min}$ ranging from 0.1 to 0.7 delivers the strongest performance while other HOT-GP variants trail behind, showing that lower levels of optimism result in poorer performance. This disparity is likely due to the substantial sparsity in rewards and difficult maze layout, characterized by multiple dead ends, which necessitates

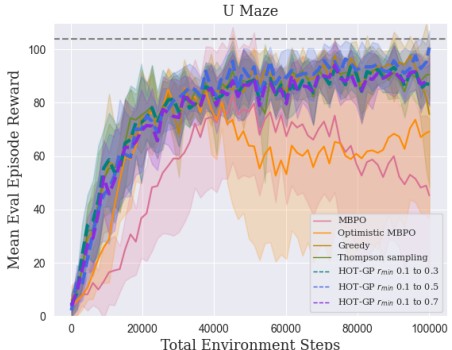 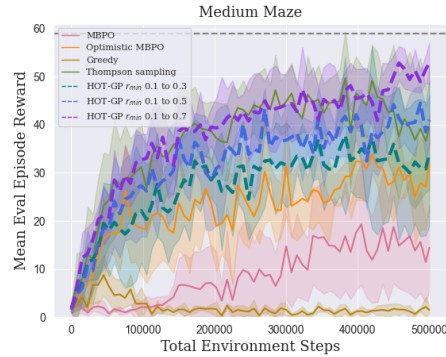

Figure 2: Learning curves on sparse maze tasks averaged over 10 seeds on the U Maze and 5 seeds on the Medium Maze. HOT-GP achieves equivalent or superior sample efficiency and performance on both tasks. The dashed line denotes SAC performance after 2,000,000 environment steps.

more aggressive optimistic exploration. Thompson sampling initially matches the best HOT-GP performance, but later shows instability with a performance decline that may be due to stochastic next state sampling. In contrast, MBPO and optimistic MBPO learn less efficiently, further reinforcing HOT-GP's advantage in jointly modeling reward and state distributions for effective exploration.

## 6.2 ROBOT COVERAGE

**Coverage Description.** We consider a practically motivated continuous control robotics problem where an agent must visit as many unique, high valued cells as possible within a finite episode. This mirrors real-world target-tracking tasks where an agent monitors multiple regions of interest under time constraints (Robin & Lacroix, 2016; Xin et al., 2024). The reward for visiting a new 2D cell is distributed as the probability density function of a mixture of three randomly placed Gaussians over the $x$ and $y$ axes, so there are cluster rewards that the agent must find and exploit. To incentivize thorough coverage, the agent receives no reward for revisiting cells in an episode.

**Comparison Methods.** As in the MuJoCo tasks, we evaluate HOT-GP with three levels of optimism and compare against Thompson sampling and greedy exploitation. A distinction is that, since DDPG outperforms SAC in this task, we use DDPG with a probabilistic actor for the `PolicySearch` algorithm. Learning a stochastic policy allows for a baseline level of exploration, which we found to be helpful when also incorporating uncertainty induced by the reward-dynamics model. For model-free baselines, we use DDPG, SAC, and the on-policy PPO algorithm (Schulman et al., 2017). For an optimistic baseline, we use two variations of H-UCRL: one has privileged access to the ground truth reward function $f_r$ while the other adapts to our unknown reward setting by learning the reward function $\hat{f}_r$ and applying it as in equation 6. In both cases, we use the same dynamics-reward model with H-UCRL as in HOT-GP for a fair comparison of the optimistic exploration strategies.

**Performance Analysis.** The first two subfigures in Figure 3 show the training curves for these approaches with and without access to the ground-truth reward function. Our method outperforms the model-based approaches in both settings and achieves DDPG-best performance marginally ahead of DDPG. The poorer performance of greedy exploitation without access to the true reward function suggests that strategic exploration is key to efficient learning for this task. Indeed, the agent must explore the environment sufficiently well enough to learn the relationship between its observations and the locations of the Gaussian centers, make informed decisions about which Gaussian center to prioritize, and realize the importance of avoiding previously seen cells. HOT-GP also demonstrates performance superior to H-UCRL, which does not converge to the optimal performance level within 200,000 environment interactions, with and without access to the ground truth reward. This may be due to infeasible optimism where the rewards associated with perturbed states $\mu(s, a) + \beta \Sigma(s, a)\eta$ are high but the states are not reachable in practice. HOT-GP avoids this problem by selecting states conditioned on optimistic rewards from the joint distribution, thus ensuring the states are plausible under the learned model and reward value. These findings reinforce the benefits of principled optimistic exploration for improving sample efficiency in tasks where exploration is key.

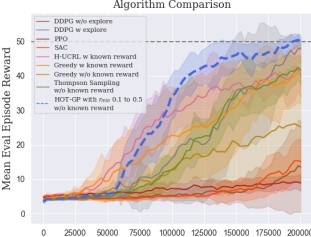 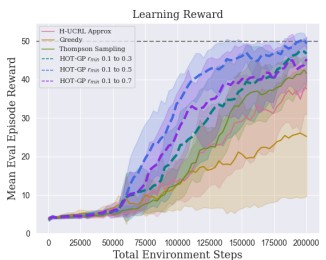 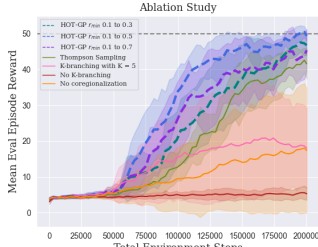

Figure 3: Learning curves in the coverage environment averaged over 10 seeds. HOT-GP achieves strong performance within 200,000 environment steps while other methods exhibit poorer sample efficiency or asymptotic performance. The dashed line denotes DDPG performance at convergence within 500,000 environment steps.

**Ablation Study.** Finally, we conduct an ablation study over the main components of HOT-GP to investigate their relative importance. The ablation study in Figure 3 presents the findings. We observe that scaling $r_{\min}$ from $0.1$ to $0.5$ yields the best results, indicating that this task benefits from a medium level of optimistic exploration. Since HOT-GP using other ranges of $r_{\min}$ values still performs fairly well, our approach does not appear to be overly sensitive to $r_{\min}$ settings. Indeed, $r_{\min}$ is an interpretable parameter that can be informed by the hypothesized level of exploration required. The ablation study in Figure 3 shows that HOT-GP without $k$-branching results in consistently poor performance. We also observe that $k$-branching with $k = 5$ leads to less optimal performance than with $k = 1$. This aligns with prior findings that accumulating errors in the dynamics model can be detrimental to learning (Janner et al., 2019; Gu et al., 2016). Furthermore, this highlights the importance of learning a faithful model to benefit from optimistic exploration, as optimism guided by an inaccurate belief harms learning. The ablation study also reveals the importance of modeling the correlation between the predicted state and reward, as this is also essential to be able to harness optimistic exploration. Lastly, sampling the next state from an optimistic distribution with Thompson sampling leads to a larger variance in performance than selecting the expected next state, thus validating our two-step sampling approach. Overall, this ablation study confirms that each component of HOT-GP is necessary to learn effectively.

## 7 CONCLUSION

We have introduced a principled approach to optimistic exploration based on leveraging a joint belief over the reward and state distribution. Our proposed algorithm, HOT-GP, uses a Gaussian Process to approximate the reward-dynamics and allows us to simulate plausible transitions associated with optimistic rewards, in a practical adaptation of Thompson sampling. Our experiments showed that HOT-GP achieves comparable or superior sample efficiency relative to other model-based methods and exploration strategies. Notably, we found that joint model uncertainty over outputs is crucial for effective exploration, particularly in challenging settings with sparse rewards, action penalties, and difficult-to-explore regions. Ultimately, this work establishes the importance of joint uncertainty modeling for optimistic exploration and makes a meaningful advancement towards more sample-efficient reinforcement learning.

**Future Work.** Future work should study how the schedule on $r_{\min}$ affects optimism and principled ways for setting this value throughout training, as we only considered a linear schedule. Another promising avenue would extend the planning horizon. We concentrated on learning 1-step looka-head reward-dynamics as HOT-GP can implicitly propagate uncertainty across the horizon by using pointwise uncertainty estimates from the model. Nevertheless, there may be benefits in explicitly propagating uncertainty to facilitate optimism over longer-term reward predictions (Seyde et al., 2021; Mehta et al., 2022; Hansen et al., 2024). Furthermore, our investigation indicates that the utility of optimistic exploration depends on the complexity of the task, although our experiments were limited to six tasks. Future research should explore the role of optimism across a broader range of problems, including in real-world scenarios with on-robot learning.

## ACKNOWLEDGMENTS

J. Bayrooti is supported by a DeepMind scholarship. A. Prorok is supported in part by European Research Council (ERC) Project 949940 (gAIa). We also sincerely thank Markus Wulfmeier and Aidan Scannell for insightful conversations and valuable feedback, which significantly contributed to the development of this work.

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

# A ALGORITHM DETAILS

In this section, we provide more detailed information about the algorithms we use for evaluation. Note that the code we used to implement these algorithms is available at `https://github.com/jbayrooti/hot_gp`.

## A.1 MODEL-FREE BASELINES

We compare model-based RL approaches with the following popular model-free algorithms:

**SAC.** We use a standard version of SAC as introduced in Haarnoja et al. (2018). Our implementation utilizes a Gaussian policy and automatic entropy tuning.

**PPO.** We use a standard version of PPO as introduced in Schulman et al. (2017).

**DDPG.** We use a variant of DDPG (Lillicrap et al., 2015) with a probabilistic actor network. In standard DDPG, the deterministic nature of the policy can lead to insufficient exploration of the action space. We mitigate this tendency by using a probabilistic actor, which incorporates exploration into the policy by sampling actions from a probability distribution. Specifically, the probabilistic actor network outputs a mean $\mu_\theta(s)$ and standard deviation $\sigma_\theta(s)$ for the current state $s$ and we select the action by sampling from the Gaussian distribution $a \sim \mathcal{N}(\mu_\theta(s), \sigma_\theta^2(s))$. In our experiments, we compare with this version of DDPG using natural exploration under the label "DDPG w/o explore". With a deterministic policy, it is also common to facilitate exploration by adding noise to the actions. Therefore, we additionally evaluate our DDPG baseline with explicit exploration under the label "DDPG w explore". Specifically, we add Gaussian noise to the sampled action with $a = a + \zeta$ for $\zeta \sim \mathcal{N}(0, \sigma_{\text{explore}}^2)$.

## A.2 GREEDY EXPLOITATION

This exploration strategy follows the optimization rule given in equation 4. In particular, we hallucinate the next state as the mean output from the model and disregard the uncertainty around the prediction. In one variation, we assume access to the ground-truth reward function $f_r$ (referred to as "Greedy w known reward") for a comparison with H-UCRL. In another variation, we also learn the reward function (referred to as "Greedy w/o known reward" in the coverage experiment and "Greedy" in the MuJoCo experiments) and similarly hallucinate rewards as the mean output from the model. In pseudocode, this approach follows Algorithm 1 exactly with the following algorithm-specific routine replacing line 10 to select the next state and reward:

- Compute the next expected state as $\hat{s}_{k+1} = \mu_s(\hat{s}_k, \hat{a}_k)$
- Observe the expected reward $\hat{r}_k = \mu_r(\hat{s}_k, \hat{a}_k)$ or true reward $\hat{r}_k = f_r(\hat{s}_k, \hat{a}_k)$

## A.3 MBPO

We implement MBPO as described in Janner et al. (2019) and summarized in Section 6.1. Specifically, the model is an ensemble of probabilistic neural networks that each predict a mean output and variance (exponentiated log variance in practice) $\mu^i(s_k, a_k)$ and $\Sigma^i(s_k, a_k)$ and parameterize a multivariate Gaussian distribution with diagonal covariance $p^i(s_{k+1}, r_k \mid s_k, a_k) = \mathcal{N}(\mu^i(s_k, a_k), \Sigma^i(s_k, a_k))$. To predict an output, the parameters are bootstrapped using expectation or randomly selecting a member and the outcome is drawn from the Gaussian distribution. This is described with the following steps replacing line 10 in Algorithm 1:

- Compute the predictions for each ensemble member $\mu^i(\hat{s}_k, \hat{a}_k), \Sigma^i(\hat{s}_k, \hat{a}_k) \leftarrow \hat{f}^i(\hat{s}_k, \hat{a}_k)$
- Bootstrap predictions to get $\bar{\mu}(\hat{s}_k, \hat{a}_k), \bar{\Sigma}(\hat{s}_k, \hat{a}_k)$
- Sample outcome from the Gaussian distribution $\hat{s}_{k+1}, \hat{r}_k \sim \mathcal{N}(\bar{\mu}(\hat{s}_k, \hat{a}_k), \bar{\Sigma}(\hat{s}_k, \hat{a}_k))$

For the optimistic MBPO baseline, we maintain the ensemble model structure but perform a procedure similar to HOT-GP, following these steps to generate next states and rewards:

- Compute the predictions for each ensemble member $\mu^i(\hat{s}_k, \hat{a}_k), \Sigma^i(\hat{s}_k, \hat{a}_k) \leftarrow \hat{f}^i(\hat{s}_k, \hat{a}_k)$

- Bootstrap predictions to get $\bar{\mu}(\hat{s}_k, \hat{a}_k), \bar{\Sigma}(\hat{s}_k, \hat{a}_k)$
- Separate the reward distribution by taking its component
  $p(s_{k+1}, r_k \mid \hat{s}_k, \hat{a}_k)_{[r]} = \mathcal{N}(\bar{\mu}(\hat{s}_k, \hat{a}_k), \bar{\Sigma}(\hat{s}_k, \hat{a}_k))_{[r]}$
- Sample reward from an optimistic reward distribution
  $\hat{r}_k \sim p(s_{k+1}, r_k \mid \hat{s}_k, \hat{a}_k, \hat{r}_k > r_{\min})_{[r]}$
- Select expected next state from state distribution $\hat{s}_{k+1} = \mathbb{E}_{p(s_{k+1} \mid \hat{s}_k, \hat{a}_k)_{[:r]}}[s_{k+1}]$

## A.4 H-UCRL

We implement H-UCRL (Curi et al., 2020) using the sampling procedure introduced in (Sessa et al., 2022). On each model-generated rollout step, this approach hallucinates $Z$ candidate next states that are plausible under the learned dynamics model and selects the one leading to the highest reward under $f_r$. Therefore, we assume access to the ground-truth reward function $f_r$ and only learn the dynamics model $f_t$ in our implementation of H-UCRL. In pseudocode, this approach follows Algorithm 1 exactly with the following steps replacing line 10 where we refer to $\sigma_s(s, a)$ as the standard deviation of an output from the GP dynamics model:

- Draw random perturbations $\eta^j \sim \text{Uniform}([-1, 1]^p)$ for $j = 1, \dots, Z$
- Compute plausible next states $\hat{s}_{k+1}^j = \mu_s(\hat{s}_k, \hat{a}_k) + \beta \, \sigma_s(\hat{s}_k, \hat{a}_k)\eta^j$
- Select next state $\hat{s}_{k+1} = \arg\max_{\hat{s}_{k+1}^j} f_r(\hat{s}_{k+1}^j, \hat{a}_{k+1}^j)$ for associated actions $\hat{a}_{k+1}^j$
- Observe the true reward $\hat{r}_k = f_r(\hat{s}_k, \hat{a}_k)$

We also adapt H-UCRL to our setting where the reward is unknown with a variation we call "H-UCRL Approx". We do this by learning both the dynamics and reward models and selecting the candidate state leading to the highest *predicted* reward, i.e., under $\hat{f}_r$ rather than $f_r$.

## A.5 THOMPSON SAMPLING

This approach follows the optimization rule given in equation 5. Our Thompson Sampling implementation only differs from HOT-GP in that we sample both the reward and next state from the optimistic reward-dynamics distribution. In practice, we implement this as described in Algorithm 1 with the following algorithm-specific routine replacing line 10:

- Compute predictive posterior distribution $p(s_{k+1}, r_k \mid \hat{s}_k, \hat{a}_k)$ using $p(f \mid \mathcal{D}_{\text{env}})$
- Sample an optimistic reward $\hat{r}_k \sim p(r_k \mid \hat{s}_k, \hat{a}_k, r_k > r_{\min})$
- Sample a plausible next state $\hat{s}_{k+1} \sim p(s_{k+1} \mid \hat{s}_k, \hat{a}_k, r_k = \hat{r}_k)$

## A.6 HOT-GP

In HOT-GP, we sample an optimistic reward and select the most likely next state associated with the sampled reward on each step of the model-generated rollout, as described in equation 9. The steps are detailed in Algorithm 1 with the following routine replacing line 10:

- Compute predictive posterior distribution $p(s_{k+1}, r_k \mid \hat{s}_k, \hat{a}_k)$ using $p(f \mid \mathcal{D}_{\text{env}})$
- Sample an optimistic reward $\hat{r}_k \sim p(r_k \mid \hat{s}_k, \hat{a}_k, r_k > r_{\min})$
- Select expected next state $\hat{s}_{k+1} = \mathbb{E}_{p(s_{k+1} \mid \hat{s}_k, \hat{a}_k, r_k = \hat{r}_k)}[s_{k+1}]$

## B TRAINING

**Learning Framework.** We implement our learning framework with MBRL-Lib (Pineda et al., 2021) for the MuJoCo tasks and TorchRL (Bou et al., 2024) for the coverage task. Additionally, we use GPyTorch (Gardner et al., 2018) to build the GP reward-dynamics model.

Table 1: Task-specific hyperparameter values

| Hyperparameter | Half-Cheetah | Reacher | Pusher | Sparse Reacher | Coverage |
|---|---|---|---|---|---|
| Environment steps $N$ | 250,000 | 20,000 | 20,000 | 37,500 | 200,000 |
| Steps per rollout $T$ | 1000 | 150 | 150 | 150 | 150 |
| Model rollouts $M$ | adaptive | adaptive | adaptive | adaptive | 150 |
| Model rollout steps $K$ | 1 | 1 | 1 | 1 | 1 |
| Batch size $B$ | 256 | 256 | 256 | 256 | 150 |
| Discount factor $\gamma$ | 0.99 | 0.99 | 0.99 | 0.99 | 0.9 |
| Learning rate $\alpha$ | 0.001 | 0.001 | 0.001 | 0.001 | 0.00005 |
| Mixing factor $\tau$ | 0.005 | 0.005 | 0.005 | 0.005 | 0.005 |
| Replay buffer size | unlimited | unlimited | unlimited | unlimited | 20,000 |

**Gaussian Process with Neural Network Mean Function.** A standard Gaussian Process is defined by a mean function $m(x)$ and a covariance function $k(x, x')$, where $x$ and $x'$ are input vectors, and $m(x)$ is typically assumed to be a zero mean. In this work, we replace the traditional zero mean function with a neural network so that $m(x) = f_\theta(\mathbf{x})$, where $\theta$ are the parameters of the network. Given inputs $X = [x_1, \ldots, x_n]$ and outputs $y = [y_1, \ldots, y_n]$, the posterior distribution of this GP is

$$p(y|X, y) = \mathcal{N}(m, K + \sigma^2 I),$$

where $m = [f_\theta(x_1), \ldots, f_\theta(x_n)]$ is the mean vector of neural network outputs for each input, K is the kernel matrix $K_{ij} = k(x_i, x_j)$, $\sigma^2$ is the noise variance, and $I$ is the identity matrix. To optimize this model, we follow the procedure in Iwata & Ghahramani (2017). This entails optimizing $\theta$ using the Mean Squared Error loss and then optimizing the kernel hyperparameters and noise variance $\sigma^2$ on the train dataset with transformed targets $y - m$, as described in the following section, on each iteration. For predicting on a new input $x_*$, the predictive posterior distribution is $p(y_* \mid x_*, X, y) = \mathcal{N}(y_* \mid u_*, \sigma_*)$ with:

$$\mu_* = m_* + K_*^T \left(K + \sigma^2 I\right)^{-1} (y - m)$$

$$\sigma_*^2 = K_{**} - K_*^T \left(K + \sigma^2 I\right)^{-1} K_*,$$

where $m_* = f_\theta(x_*)$, $\mathbf{K}_* = [k(x_*, x_1), \ldots, k(x_*, x_n)]^T$, $K_{**} = k(x_*, x_*)$. Thus, the neural network learns the non-zero mean function while the GP covariance function is modeled by the kernel as usual, thus allowing the model to quantify uncertainty around complex functions.

**Reward-Dynamics Model Training.** Our approach requires access to the full posterior distribution of the Gaussian process. To facilitate this in an efficient manner, we train the GP using a variational bound based on inducing points (Hensman et al., 2015) following the approach given in Iwata & Ghahramani (2017). This allows for efficient training using stochastic inference. To further reduce the computational cost of learning the model, we do not train on the whole dataset on each iteration and instead draw $1,000$ randomly sampled transitions from $\mathcal{D}_{\text{env}}$. This approach reduces the training cost significantly and provides an additional source of stochasticity that helps training. Using a Gaussian likelihood allows us to compute the predictive posterior in closed form, allowing us direct access to its mean and variance.

**Hyperparameters.** We train all algorithms using a set of hyperparameters that we found to be optimal through trial-and-error or were recommended. Table 1 describes the hyperparameter values specific to the MuJoCo and VMAS tasks we considered. We also report additional algorithm-specific hyperparameters used. In H-UCRL, we use the exploration-exploitation coefficient $\beta = 0.01$ and number of samples $Z = 5$. In MBPO and optimistic MBPO, we use an ensemble of size 7. For DDPG, we use an exploration noise $\sigma_{\text{explore}}$ from 1 to 0.1. For GP-based approaches, we use the Matern kernel as the covariance function and learn from 100 inducing points. For the MuJoCo tasks, we use a 4-layer MLP with 200 hidden units per layer and SiLU activation function. For the VMAS coverage task we use a 2-layer MLP with 200 hidden units per layer and Mish activation function. We use the Adam optimizer in all cases. For further information on our implementation, please see our open-sourced implementation at this repository:
`https://github.com/jbayrooti/hot_gp`.

Table 2: Computation times in hours across tasks and algorithms

| Wall Clock Runtime | MBPO | Optimistic MBPO | Greedy | HOT-GP |
|---|---|---|---|---|
| Half-Cheetah | $24.5 \pm 3.5$ | $26.7 \pm 4.88$ | $18.5 \pm 3$ | $28.89 \pm 4.1$ |
| Reacher | $1.89 \pm 0.1$ | $2.04 \pm 0.2$ | $2 \pm 0.1$ | $5.18 \pm 0.2$ |
| Pusher | $1.87 \pm 0.2$ | $2.05 \pm 0.2$ | $1.92 \pm 0.4$ | $6.53 \pm 0.1$ |
| Sparse Reacher | $3.76 \pm 0.3$ | $4.1 \pm 0.4$ | $3.94 \pm 0.3$ | $8.55 \pm 1.5$ |
| U Maze | $12.45 \pm 1.1$ | $12.43 \pm 1.7$ | $11.75 \pm 1.6$ | $21.08 \pm 3.8$ |

Figure 4: Frames from a rollout of a HOT-GP trained policy in the coverage environment. The agent (purple circle) drives around to cover target areas which are brightly colored. The target locations change on each episode.

**Computation Time Comparison.** We report the mean wall clock runtime and standard deviations for a single run of each algorithm on the MuJoCo tasks considered in Table 2. These results are averaged over 10 seeds. Our HOT-GP implementation incurs longer wall clock runtimes due to computational cost of GP training. However, recent research proposes promising techniques that can mitigate this overhead (Wenger et al., 2024; Chang et al., 2023). Also, note that our approach is not tied to GP models and can seamlessly generalize to alternative models such as Bayesian Neural Networks (BNNs), as the model is only needed to approximate the joint posterior distribution over next states and rewards.

## C  EVALUATION TASKS

In this work, we evaluate HOT-GP on five continuous control tasks: four using the MuJoCo physics engine and one using the VMAS simulator. In this section, we outline the setup for each task.

**Half-Cheetah.** The Half-Cheetah is a MuJoCo continuous control task where a 2D bipedal agent, resembling a simplified cheetah, must learn to run as quickly as possible. The observations are 17-dimensional vectors that include the position and velocity of the agent's torso and 6 joints. Actions are 6-dimensional vectors, where each component corresponds to the torque applied to one of the agent's joints. The reward is primarily based on the forward velocity of the agent's torso while also penalizing excessive torques on the joints. Let $x_0$ be the agent's original position, $x_1$ be the position after a simulation step, and $dt$ be the time between. Then the reward is given by $r(s, a) = \frac{x_1 - x_0}{dt} - 0.1 \sum_{i=1}^{6} a_i^2$.

**Pusher.** The Pusher task in MuJoCo is a continuous control task where a robotic arm must learn to push an object to a target location on a 2D plane. The agent controls a three-link planar arm and must manipulate an object to move it towards a fixed goal position. The observation space is a 23-dimensional vector, which includes the position and velocity of the robot arm's joints, position and velocity of the object being pushed, and the position of the target. The ac-

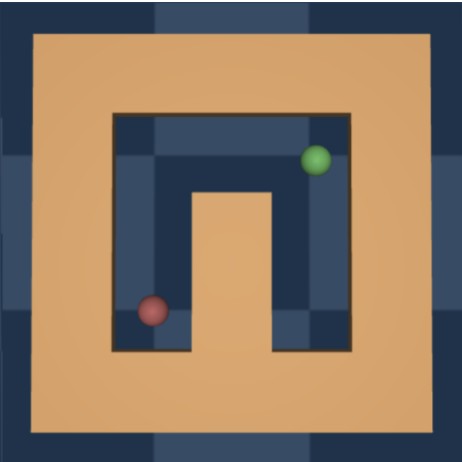 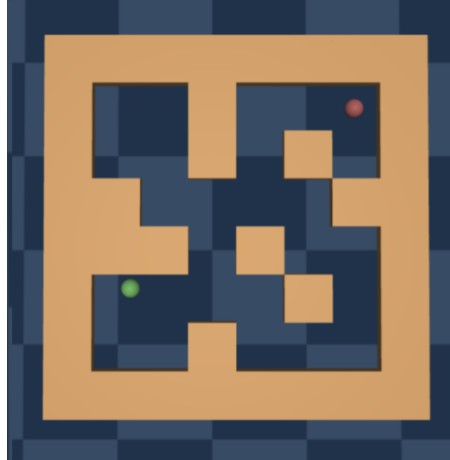

Figure 5: Visualizations of the U Maze (left) and Medium Maze (right) environments with randomly selected goal locations.

tion space is a 7-dimensional vector, where each component represents the torque applied to one of the joints in the robotic arm. The reward function is based on the Euclidean distance between the object and the target location, incentivizing the agent to move the object closer to the target while also penalizing large torques. Let $p_{\text{obj}}$ be the position of the object, $p_{\text{goal}}$ be the position of the goal, $p_{\text{arm}}$ be the position of the arm's tip. Then the reward is given by

$r(s, a) = -\|p_{\text{obj}} - p_{\text{goal}}\|_2 + 0.1 \cdot \left(-\sum_{i=1}^{7} a_i^2\right) + 0.5 \cdot (-\|p_{\text{obj}} - p_{\text{arm}}\|_2)$.

**Reacher.** The Reacher task in MuJoCo is a continuous control environment where a robotic arm must learn to position its end effector at a goal location in 3D space. The observations are 19-dimensional vectors that include the joint angles and velocities of the robotic arm. Actions are 6-dimensional vectors, where each component corresponds to the torque applied at one of the arm's joints. The reward incentivizes the agent to reach the goal while also penalizing excessive control actions. Letting $EE_{\text{pos}}(s)$ represent the position of the end effector and $\rho$ be the action penalty, the reward function is $r(s, a) = -\sum_{i=1}^{3}(EE_{\text{pos}}(s)_i - \text{goal}_i)^2 - \rho \cdot \|a\|^2$.

**Sparse Reacher.** The Sparse Reacher is the same as the Reacher task, except with a modified reward structure. Let $EE(s)$ represent the position of the end effector, $\epsilon$ be a small threshold distance, and $\rho$ be the action penalty. The reward function is

$$r(s, a) = -\rho\left(\exp\left(-\sum_{i=1}^{6} a_i^2\right) - 1\right) + \begin{cases} \exp\left(-\|EE(s) - \text{goal}\|^2\right) & \text{if } \|EE(s) - \text{goal}\| < \epsilon \\ 0 & \text{otherwise.} \end{cases}$$

The first term rewards the agent based on the negative squared distance between the end effector's position and the target goal if the distance is less than $\epsilon$ and gives no reward otherwise. The second term penalizes the agent for applying excessive torques to the joints. This design creates a challenge for exploration as the agent receives minimal feedback about its performance until it is sufficiently close to the target. We take inspiration for this setup from Curi et al. (2020) although our rewards are more sparse and we use $\epsilon = 0.2$ in our experiments.

**U Maze.** In this task, the agent must navigate through a U-shaped maze shown in Figure 5 to reach a randomized target location. The agent receives a sparse reward of 1 when it comes within a small threshold distance $\epsilon$ of the target and 0 otherwise. Specifically, the reward function is

$$r(s, a) = \begin{cases} 1 & \text{if } \|\text{Pos}(s) - \text{goal}\| < \epsilon \\ 0 & \text{otherwise.} \end{cases}$$

The agent observes its position $(x, y)$ and 2-dimensional goal position. Actions are 2-dimensional vectors representing velocity in the $x$ and $y$ dimensions.

**Medium Maze.** This task shares the same observation and reward structure with the U Maze. The only difference is that this maze environment is significantly more complex, as shown in Figure 5.

**Coverage.** In this VMAS task, the agent must discover and visit unique, discretized cells with high value in a continuous state-action space. The reward for visiting a new cell at coordinates $(x, y)$ is distributed as the probability density function of a mixture of three Gaussians over the $x$ and $y$ axes,

$$p(x, y) = \frac{1}{3} \sum_{i=1}^{3} \mathcal{N}((x, y) \mid (\mu_i, \Sigma)). \tag{10}$$

To incentivize thorough coverage of the regions around each Gaussian center, the agent receives no reward for revisiting cells in an episode. The agent observes its position $(x, y)$, velocity, current value $p(x, y)$, values of neighboring cells $p(x_n, y_n)$ for $(x_n, y_n) \in \mathcal{N}(x, y)$, and the target locations $\{\mu_i\}_{i=1}^{3}$. The target locations are randomly generated in each episode so that the agent must learn the high-level pattern of reward distribution as a function of $(x, y)$ and $\{\mu_i\}_{i=1}^{3}$, and $\Sigma$. Throughout our experiments, we use $\Sigma = 0.05$. We provide snapshots from sample rollouts on the coverage task in Figure 4.

# D    EXTENDED EXPERIMENTS

In this section, we extend our experiments in order to better contextualize HOT-GP. First, we investigate the asymptotic performance of HOT-GP and all comparison methods on the Sparse Reacher task. We also provide learning curves on the Sparse Reacher task with action penalty 0.3 to supplement the results in Figure 1) with an intermediate action penalty. Finally, we evaluate the approximation of H-UCRL and MBPO without uncertainty clamping on the Half-Cheetah task.

**Sparse Reacher.** To supplement the Sparse Reacher results presented in Figure 1, we provide an additional learning curve on the Sparse Reacher using action penalty of 0.3. These results are consistent with the trends observed using other action penalties and HOT-GP variants perform the best.

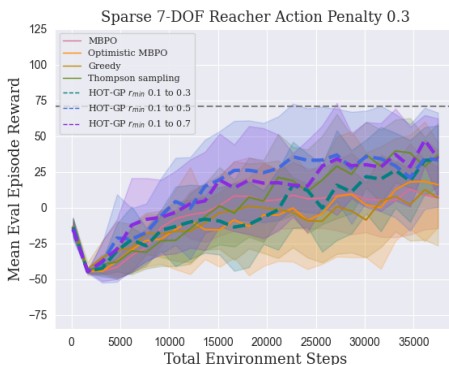
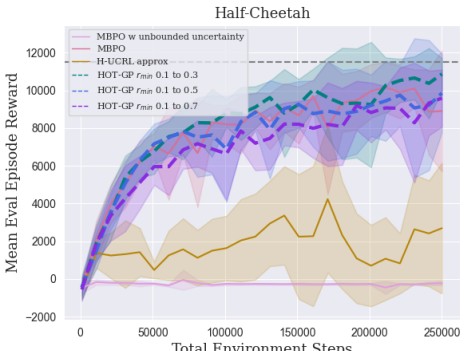

Figure 6: Learning curves on the Sparse Reacher with action penalty of 0.3. HOT-GP achieves superior sample efficiency and performance. The dashed line denotes SAC performance at convergence within 1,000,000 environment steps.

Figure 7: Learning curves for select methods on Half-Cheetah averaged over 5 seeds. The dashed line denotes SAC performance at convergence within 3,000,000 environment steps for Half-Cheetah.

**Half-Cheetah.** As described in Section A, H-UCRL approx differs from the algorithm presented in Curi et al. (2020) in a few ways: (1) we use our GP model rather than a probabilistic ensemble, (2) we use the sampling-based optimistic hallucination method in equation 6 to select states, and (3) we use predicted rewards to guide optimism rather than the ground truth rewards. This version of H-UCRL does not perform well on Half-Cheetah relative to MBPO and HOT-GP, with the best seeds achieving mean evaluation episode rewards as high as 6000 after training with 250,000 environment steps. As we observed when discussing H-UCRL performance on the coverage task, this performance gap may stem from infeasible optimism, where the rewards associated with perturbed states $\mu(s, a) + \beta \Sigma(s, a)\eta$ are high but unreachable in practice. Additionally, a suboptimal choice of $\beta$ could be a contributing factor, as we did not fine tune this parameter on Half-Cheetah due to

its extended runtime and instead used the optimal $\beta$ finetuned on the coverage task. HOT-GP also has a hyperparameter $r_{\min}$, which controls the level of optimism during training. While the domain dependence of $r_{\min}$ means some adaptation is required for new tasks, we found that setting $r_{\min}$ is generally intuitive and, moreover, provides interesting insights into the nature of the task itself. For instance, if the dynamics in the domain are simple, we can be more optimistic about exploration (i.e., use a higher value of $r_{\min}$), whereas if they are complex, we need to spend more time learning the dynamics before becoming more optimistic.

We also experimented with removing uncertainty clamping from MBPO, which resulted in a complete failure to learn as shown in Figure 7. Standard MBPO uses an ensemble of probabilistic neural networks that predicts both the mean and log variance for a given input, with log variances clamped to prevent excessive growth. Removing this clamping significantly degrades performance, indicating that the model uncertainties are poorly calibrated and heavily dependent on clamping to maintain stable learning. This finding reinforces our choice of model, as GPs provide well-calibrated uncertainties (Chowdhury & Gopalan, 2017).

