# OpenReview forum: "Efficient Model-Based Reinforcement Learning Through Optimistic Thompson Sampling"
_ICLR.cc/2025/Conference — ICLR 2025 Poster_

### Official Review · Reviewer_LvwR · 2024-10-31

**Soundness:** 2
**Presentation:** 2
**Contribution:** 2
**Rating:** 5
**Confidence:** 4

**Summary:**

The paper proposes a model-based reinforcement learning algorithm called HOT-GP (Hallucination-based Optimistic Thompson sampling with Gaussian Processes). This approach leverages Gaussian Processes to jointly model the reward and transition dynamics, addressing both state and reward uncertainties. The main goal is to optimize exploration by focusing on high-reward regions in the state-action space. HOT-GP introduces a new optimism-driven Thompson sampling strategy to balance exploration and exploitation, which is particularly useful in environments with sparse rewards or penalties for actions.

**Strengths:**

The HOT-GP proposed in the paper performs joint uncertainty modeling between rewards and state dynamics. This method innovatively combines the uncertainties in rewards and state transitions, achieving more efficient exploration in environments with sparse rewards and high penalties.

**Weaknesses:**

1. The experiments are insufficient. For environments with sparse rewards and difficult exploration, tasks like Maze and Navigation, such as Sparse-Point-Robot, might better illustrate the effectiveness of the approach.
2. In some environments, the training curves presented in the paper do not appear to have converged, leaving the convergence of the algorithm unclear and making it difficult to judge whether an optimal policy has been achieved.
3. The paper seems to lack ablation studies on different modules and tasks, which fails to adequately demonstrate the independent contributions of various components to overall performance. Without detailed ablation studies, it becomes challenging to evaluate the individual effects of joint uncertainty modeling and the optimistic strategy across different tasks.

**Questions:**

According to the paper, the HOT-GP algorithm is inspired by H-UCRL. Why, then, was H-UCRL not used as a comparative algorithm in the MuJoCo tasks?

---

> ### Author Response · Authors · 2024-11-20
> **Response by Authors**
>
> Thank you very much for taking the time to review our paper. We appreciate your comments and suggestions to improve our evaluation. Please find our responses below.
>
> **_Weaknesses_**
>
> 1. The experiments are insufficient. For environments with sparse rewards and difficult exploration, tasks like Maze and Navigation, such as Sparse-Point-Robot, might better illustrate the effectiveness of the approach.
>
> **Response:** Thank you for the recommendation to consider Sparse Point Maze environments. We have taken your suggestion to evaluate HOT-GP and baseline algorithms on two additional Sparse Point Maze environments: U Maze and Medium Maze. Currently, we have finished gathering the results on U Maze and are still running experiments on the Medium Maze. Our results on the U Maze are consistent with our findings on the Sparse Reacher task and demonstrate the superior sample efficiency of HOT-GP in sparse reward settings. Furthermore, these results highlight the stability of our approach against MBPO and optimistic MBPO.
>
> Additionally, we have strengthened our experiments by increasing the number of seeds to 10 and evaluated on the Sparse Reacher task with an intermediate action penalty of 0.3. We have uploaded a new paper pdf with all these new results except for the Medium Maze, which will be included when the experiments finish.
>
> 2. In some environments, the training curves presented in the paper do not appear to have converged, leaving the convergence of the algorithm unclear and making it difficult to judge whether an optimal policy has been achieved.
>
> **Response:** While it is true that some training curves do not converge within the maximum number of environment steps, this is secondary to the main motivation of our evaluation. Our primary objective is to evaluate the sample efficiency of HOT-GP relative to existing approaches by comparing algorithms’ performance after a fixed number of environment interactions, as done in H-UCRL (and many other papers). The question of whether algorithms converge within the set number of steps is less relevant and involves balancing computational costs across multiple tasks while ensuring useful results. We have chosen the maximum number of environment steps that allows most algorithms to converge whenever possible, given our computational limitations. It is important to note that, in all experiments apart from the Sparse Reacher, at least one HOT-GP variant (of the three considered $r_{\text{min}}$ schedules) achieves the SAC-best performance after training for a fraction of the environment steps.
>
> 3. The paper seems to lack ablation studies on different modules and tasks, which fails to adequately demonstrate the independent contributions of various components to overall performance. Without detailed ablation studies, it becomes challenging to evaluate the individual effects of joint uncertainty modeling and the optimistic strategy across different tasks.
>
> **Response:** We would like to point out that our paper includes an ablation study in Figure 3 (the old Figure 2) and is discussed in Section 6.2. In this study, we ablate over values for the $r_{\text{min}}$ threshold ($r_{\text{min}}$ ranges from 01 to 0.3, 0.5, and 0.7) and the number of steps K in a model rollout (K = 1, 5, T). We also consider HOT-GP with component-wise independent predictions (no coregionalization) and HOT-GP with sampling the next state (Thompson sampling) rather than selecting the expected next state. Our ablation study provides insights into the importance of each element of our proposed approach. Please let us know if there are specific ablations that you believe are missing.
>
> **_Questions_**
>
> 1. According to the paper, the HOT-GP algorithm is inspired by H-UCRL. Why, then, was H-UCRL not used as a comparative algorithm in the MuJoCo tasks?
>
> **Response:** This is a good question. While we are inspired by the notion of optimistic exploration that is central to H-UCRL, our work considers a different setting where the reward function is unknown. Specifically, H-UCRL assumes access to the ground truth reward function, making it not directly comparable with HOT-GP. One way to circumvent this problem is to estimate the reward function and perform H-UCRL using the learned reward as a surrogate for the ground truth. We tested this approximation of H-UCRL on the MuJoCo Half-Cheetah task and observed subpar performance relative to HOT-GP and MBPO, as shown in Figure 5 in Appendix D. Since we do not have access to the true reward function during training, it remains unclear whether the poor performance is due to the algorithm itself or errors in reward estimation. This uncertainty complicates a fair comparison to H-UCRL, so we opted not to do so.

---

> ### Comment · Reviewer_LvwR · 2024-11-25
>
> Theoretically, Thompson sampling has a much lower sampling efficiency than Bayes-optimal. The policies learned within the BAMDP (Bayes-adaptive Markov decision processes) framework are closer to the optimal exploration strategy. However, the methods compared by the authors did not include approaches based on the Bayes-optimal policy. Additionally, the authors' approach is not the first to 'reason about joint uncertainty over transitions and reward.' Please refer to [1], which uses VAE-based methods to model the transition and reward models, allowing for better learning of the agent's uncertainty about the environment. This method has already been shown to learn policies closer to the optimal compared to posterior sampling(Thompson sampling)-based methods  [2]. I believe this paper only modifies H-UCRL and compares it with some basic algorithms, with limited contributions. Thank you to the authors for the experiments, which have made the paper more thorough. However, I will maintain my score.
>
> [1]Zintgraf et al, Varibad: A very good method for bayes-adaptive deep rl via meta-learning. ICLR, 2020.
>
> [2]Rakelly et al. Efficient off-policymeta-reinforcement learning via probabilistic context variables. ICML, 2019.

---

> > ### Author Response · Authors · 2024-11-27
> > **Response by Authors**
> >
> > Thank you for your response. We appreciate the additional feedback and references to these related works.
> >
> > First, we would like to clarify that we do not mean to claim that ours is the first model that allows for reasoning about joint uncertainty over transitions and reward. Indeed, there are existing approaches that can model joint uncertainty over transitions and rewards, such as VAEs [1] or Bayesian neural networks. However, such models have primarily been used to improve agents’ knowledge of the environment and not for optimistically directing exploration. Crucially, our algorithm is the first to model and leverage this joint uncertainty for principled optimistic exploration. We are also the first to empirically demonstrate that modeling joint uncertainty over the state and reward distributions is key to principled, feasible, optimistic exploration since this allows the agent to generate plausible transitions aligned with (i.e., conditioned on) optimistic rewards. Our approach directs exploration towards high-reward regions while remaining grounded in the environment dynamics, thereby avoiding infeasible or irrelevant exploratory behaviors (to the extent that the model is correct). We have adjusted the wording in the abstract to better reflect the novelty of our contributions.
> >
> > Thank you for highlighting Bayes-Adaptive MDPs as a potential framework to consider quantifying epistemic uncertainty over performance and learning Bayes-optimal policies. The approach presented in [2] is certainly an interesting one, however that work considers a meta-learning setup where training occurs on a series of multiple related tasks. Since we focus on a single-task setting, we believe the approach described in [2] is not directly comparable with HOT-GP. More generally, we agree that Thompson sampling can face efficiency challenges and may have lower sample efficiency compared to Bayes-optimal methods, although the magnitude of this gap varies depending on the task. In [2], the authors show that Thompson sampling has lower sample efficiency than their Bayes-optimal approach on a gridworld environment. However, their findings lack formal guarantees and leave the generality of these results unverified on different problems. Thompson sampling still remains recognized as a well-grounded approach for exploration in reinforcement learning in the literature [3, 4, 5].
> >
> > Furthermore, we would like to point out that we are not performing traditional Thompson sampling but rather an optimistic variant where the reward is sampled from a truncated Gaussian distribution (line 837 in the paper) and the state is sampled from the resulting reward-conditioned Gaussian distribution (line 839 in the paper). Our approach is less prone to instability compared to standard Thompson sampling, which optimizes the policy over any sampled reward-dynamics function (Equation 5). While this approach to Thompson sampling achieves similar sample efficiency and performance as HOT-GP on some tasks, it leads to some performance degradation on the Medium Sparse Point Maze environment and is less sample efficient than HOT-GP on the VMAS Coverage task. In contrast, we do not observe any performance degradation with HOT-GP.

---

> ### Author Response · Authors · 2024-11-27
> **Response by Authors - Continued**
>
> We believe there may also be a misunderstanding of our approach and contributions. HOT-GP is _not_ a modification of H-UCRL. HOT-GP is based on Thompson sampling rather than Upper Confidence Bounds and operates in a fundamentally different manner. HOT-GP learns both the dynamics and reward functions with joint uncertainty while H-UCRL only learns the dynamics model and assumes access to the ground truth reward function. Learning the dynamics and reward functions allows us to reason about the joint uncertainty over possible outcomes and generate plausible transitions conditioned on optimistic rewards. In contrast, H-UCRL learns an uncertainty-aware model of the dynamics that captures the mean $\mu(s,a)$ and standard deviation $\Sigma(s,a)$ for each input $(s,a)$. They then select the next state $s’ = \tilde{f}(s,a) = \mu(s,a) + \beta\Sigma(s,a)\eta$ using $\beta$ as a fixed hyperparameter and hallucinated $\eta$ value that leads to the highest transition reward. This can result in infeasible optimism if $\mu(s,a) + \beta\Sigma(s,a)\eta$ is not reachable from the current state $s$. We do take high level inspiration from the motivation behind H-UCRL to direct exploration in an optimistic manner with respect to the single-step reward value. We also compare HOT-GP with a superset of the baselines in the H-UCRL paper, namely Thompson sampling and greedy exploitation, in addition to MBPO, an optimistic variant of MBPO, and model-free algorithms. We choose these comparison methods to illustrate important aspects of our approach. Specifically:
> - Greedy exploitation maximizes the expected reward and ignores uncertainty induced by the reward-dynamics model. We hypothesized this would lead to poorer performance than HOT-GP, demonstrating the importance of uncertainty-aware policy optimization.
> - Thompson sampling optimizes the policy with respect to a sampled optimistic reward-dynamics model. We expected similar performance to HOT-GP, with slightly less stability, showing the benefit of optimism only over the rewards.
> - MBPO uses an ensemble of reward-dynamics models to sample states and rewards but does not account for joint uncertainty. We anticipated poorer performance than HOT-GP, illustrating the importance of optimism and joint uncertainty for consistent hallucinations.
> - Optimistic MBPO is the same as MBPO but incorporates the optimistic reward sampling of HOT-GP. Similarly, we expected better performance than MBPO but poorer performance than HOT-GP due to the lack of joint uncertainty for consistent state-reward hallucinations.
> - Model-free algorithms serve as a baseline to contrast model-free and model-based approaches. We hypothesized these results would be less sample efficient than the model-based approaches.
>
> The primary contributions of our work are twofold: 1) the HOT-GP algorithm and 2) establishing the importance of joint uncertainty modeling for optimistic exploration. Our evaluations reveal that modeling and leveraging joint uncertainty over the state and reward distributions is crucial for effective optimistic exploration, particularly in challenging settings with sparse rewards, action penalties, and difficult-to-explore regions. Therefore, we anticipate that the HOT-GP algorithm and broader technique of optimism with joint uncertainty will be valuable tools for advancing exploration strategies and improving sample efficiency–core challenges at the heart of reinforcement learning.
>
> We hope we have addressed your most recent concerns as well as those raised in your original review. To summarize, these are the actions we have taken in response to the issues outlined in your initial feedback:
> - Updated the paper with completed results on two Sparse Point Mazes (Figure 2) as suggested
> - Ran additional experiments to show asymptotic performance on the Sparse Reacher (Figure 6), asymptotic performance was already clear on the other tasks
> - Highlighted our original ablation study results (Figure 3)
> - Provided results comparing HOT-GP with an appropriate approximation of H-UCRL (Figure 8) along with a justification for not conducting further comparisons with H-UCRL
>
> We would be grateful if you would consider increasing your score in light of these responses. If you have any further questions or comments, please do not hesitate to ask. Thank you!
>
> [1] Rakelly et al. Efficient off policy meta reinforcement learning via probabilistic context variables. ICML, 2019.\
> [2] Zintgraf et al. Varibad: A very good method for bayes-adaptive deep rl via meta-learning. ICLR, 2020.\
> [3] Moradipari et al. Improved bayesian regret bounds for thompson sampling in reinforcement learning. NeurIPS, 2023.\
> [4] Osband et al. Why is posterior sampling better than optimism in reinforcement learning?. ICML, 2017.\
> [5] Osband et al. More efficient reinforcement learning via posterior sampling. NeurIPS, 2013.\
> [6] Bubeck and Sellke. First-order bayesian regret analysis of thompson sampling. Algorithmic Learning Theory, 2020.

---

> ### Author Response · Authors · 2024-12-02
> **Follow Up by Authors**
>
> We wanted to follow up on our last response as the end of the discussion period is approaching. We believe we have addressed your feedback by (1) providing the additional experiments you suggested and showing asymptotic behavior; (2) highlighting our original ablation study; (3) explaining the relationship between our work and the referenced papers; and (4) clarifying the novelty of our approach and its relationship to H-UCRL. In light of these responses, if you find our work more convincing, we would greatly appreciate it if you would consider updating your score. Also, we would like to check if there are any remaining adjustments to the paper you feel are necessary to resolve your concerns. Thank you!

---

> > ### Comment · Reviewer_LvwR · 2024-12-03
> >
> > Thank you to the authors for the detailed clarifications. However, I still believe that the innovation of the paper is limited, and I have adjusted my score accordingly.

---

> > > ### Author Response · Authors · 2024-12-04
> > > **Response by Authors**
> > >
> > > Thank you very much for increasing your score and your engagement during the discussion period! We are glad that our clarifications have helped address some of your concerns. We respect your opinion regarding the impact of our paper's innovations. However, we would like to briefly reiterate the key novelties of our work to ensure they are fully recognized.
> > >
> > > The following is a list of our innovations:
> > > - We are the first to inform optimistic exploration by the joint uncertainty over reward and state distributions
> > > - We introduce a reliable mechanism for adaptive optimism
> > > - We propose a novel acquisition function based on optimism with respect to the reward and feasibility with respect to the dynamics
> > >
> > > Our HOT-GP algorithm operationalizes these principles. We develop a multi-output GP with a flexible neural network mean function that can represent complex dynamics and provides the full covariance matrix that captures the correlations between all the state components and reward. While these techniques have been proposed individually in prior work, to the best of our knowledge, ours is the first application to model-based reinforcement learning. We empirically demonstrate that HOT-GP achieves equivalent or superior sample efficiency and performance compared to alternative methods.
> > >
> > > We hope this summary clarifies the innovation of our contributions. Thank you again for your time and feedback during the review process.

---

### Official Review · Reviewer_GjSJ · 2024-11-01

**Soundness:** 4
**Presentation:** 4
**Contribution:** 4
**Rating:** 8
**Confidence:** 5

**Summary:**

This manuscript introduces HOT-GP, a novel and efficient approach to model-based reinforcement learning that enhances performance through a unique combination of techniques. The core innovation of HOT-GP lies in its ability to learn the joint distribution of dynamics and rewards, paired with an optimistic sampling strategy for "in-model" policy optimization. By leveraging the latest model-based reinforcement learning techniques, HOT-GP selects models for policy optimization based on optimistic sampling, which promotes exploration in high-potential regions of the policy space. Evaluated across several continuous-action benchmarks, HOT-GP demonstrates superior performance compared to both baseline methods and other state-of-the-art algorithms.

**Strengths:**

1) The use of the joint distribution of dynamics and rewards in the learning process is a clever and impactful approach, offering new perspectives in model-based reinforcement learning.
2) The implementation is robust and well-executed, showcasing attention to both theoretical and practical details.
3) The manuscript is excellently presented—clear, concise, and engaging. It's a pleasure to read and straightforward to follow.
4) The experiments are adequately designed, with thoughtful discussions that provide meaningful insights into the method's strengths and limitations.
5) Providing the code for replicating experiments is greatly appreciated, allowing for easy validation and further exploration by the community.
6) The appendix is comprehensive and well-detailed, giving me confidence that I could implement the method from scratch!

**Weaknesses:**

1) The evaluation would benefit from more complex scenarios, such as object manipulation or long-horizon tasks, which could further demonstrate the method’s robustness and adaptability.

2) While the integration of a neural network to learn the GP mean is discussed in the main text, I feel like a small paragraph or additional description in the appendix is required. I understand the general approach and can infer the authors' method, but a concise explanation would be a helpful addition for readers.

**Questions:**

None

---

> ### Author Response · Authors · 2024-11-20
> **Response by Authors**
>
> Thank you very much for taking the time to review our paper. We are so pleased to hear that you enjoyed our work and appreciate the practical recommendations. Please find our responses below.
>
> **_Weaknesses_**
>
> 1. The evaluation would benefit from more complex scenarios, such as object manipulation or long-horizon tasks, which could further demonstrate the method’s robustness and adaptability.
>
> **Response:** Thank you for this suggestion. To address this, we are running additional experiments on two Sparse Point Maze environments where the agent must navigate through a maze to a goal location. The sparse reward structure makes these long horizon tasks and our results so far validate the sample efficiency of our approach. We have uploaded a new paper pdf including the new results on one maze and will update the paper pdf again with results on the second maze once our experiments are complete.
>
> 2. While the integration of a neural network to learn the GP mean is discussed in the main text, I feel like a small paragraph or additional description in the appendix is required. I understand the general approach and can infer the authors' method, but a concise explanation would be a helpful addition for readers.
>
> **Response:** Thank you for highlighting this. We agree that a more detailed explanation of how the neural network is integrated to learn the GP mean would improve clarity. To address this, we added a paragraph on this, entitled Gaussian Process with Neural Network Mean Function, to the paper in Appendix B.

---

> > ### Comment · Reviewer_GjSJ · 2024-11-25
> >
> > Thank you for the extra experiments and explanations! I keep my score..

---

> > > ### Author Response · Authors · 2024-12-02
> > > **Response by Authors**
> > >
> > > Thank you very much again for reviewing our paper and acknowledging our work!

---

### Official Review · Reviewer_QgWe · 2024-11-01

**Soundness:** 2
**Presentation:** 3
**Contribution:** 2
**Rating:** 3
**Confidence:** 3

**Summary:**

This paper introduces a novel model-based reinforcement learning algorithm to explore effectively and improve sample efficiency. The introduced method uses a Gaussian processes to represent a posterior over transition functions and reward functions and models the (linearized) covariance of reward and successor states (the mean functions are later modeled by MLPs instead). The authors introduce an exploration method by modifying Thomposon sampling to be more optimistic: rewards are sampled with the condition of a minimum reward and successive states are taken to be the expected successor state, conditional on the same minimum reward. The authors conducted experiments that analyze the introduced algorithms empirically and show that their agents learns faster in early stages of training.

**Strengths:**

The paper is written clearly and it is easy to follow. The method adresses an important issue and from what I can see is applicable to a variety of algorithms. The experiments suggest that the proposed algorithm may aid sample efficiency, especially in early stages of training.

**Weaknesses:**

The main weakness of this paper to me is of conceptual nature. I have no good intuition for why the proposed algorithm performs efficient exploration. This is mainly due to two reasons:

1. I am not convinced the reward-uncertainty is significantly more useful than the dynamics uncertainty. As the authors describe, the de-facto standard way of formulating GPs is that output dimensions are considered independent predictions. The uncertainty of each dimension (the variance of the GP) of a GP conditioned on $X$ in a query-point $x$ is then given by $K(x,x) - K(x,X)K^-1(X,X)K(X,x)$ and depends only on the kernel choice and the data distribution. Unless one has specific knowledge about an appropriate reward-kernel, I have no intution  why the reward offers a better (or different) information gain signal than the dynamics.

2. It is unclear to me why the proposed sampling distribution is meaningful. The sampling distribution described in Eq. 8 seems to imply a risk-sensitive sampling strategy without motivating this. For example, one may easily construct MDPs, where optimal trajectories are guaranteed to contain a very small reward. Such trajectories could not be sampled under this strategy, despite being optimal. Mathematically speaking, it is unclear to me what the implications of conditioning on an impossible event is here (i.e., what if $r_t$ can't be greater than $r_{min}$). I can unfortunately not see why one would expect this algorithm to explore reliably.


Minor points:

- The dimensions of the GP definition in line 269 don't seem to add up to me. The covariance matrix, as written here, has dimensions $2 \times 2$, while the mean functions are of dimension $|\mathcal{S}| + 1$. I moreover think the input dimensions of covariance functions $K$ defined in line 271 should operate on pairs of $S\times A$, not simply $S \times A$.

- Expected successor states (as described in 5.2, line 323) don't seem sensible to me. The expectation of a high-dimensional state is not necessarily a meaningful or even existant state. It was moreover not clear to me, how to the model predictions are conditioned on a certain reward level in practice.

- Given that the authors use a GP, I suspect many readers would appreciate a table (or similar) comparing the computational burden of this approach with alternative methods. Furthermore I would be interested to know the effect of subsampling smaller batches (as described in line 850) on the quality of uncertainty estimates with a GP.

**Questions:**

1. Why do the authors aim to relieve the stationarity assumption in line 285. Since this GP models the transition dynamics of an MDP, the transition kernel should indeed be stationary. Aiming to address non-stationary MDPs, I believe, would entail significant complications for the entire learning algorithm.

2. In most experiments, the asymptotic performance of the shown algorithm seems to be lower than the base-SAC implementation in all of the environments. Is this due to the cutoff in plotting or is the asymptotic performance indeed lower in most environments? Inserting a dashed line as with SAC into the figure might clear this up.

3. Given that the authors perform k-branching rollouts, the value estimates bootstrapped at the end of model-generated rollouts should play an important role in quantifying uncertainty. Would omitting this uncertainty skew the ability of the algorithm to explore efficiently? It seems unlikely to me, that Thompson sampling maintains it's theoretical guarantees if one only samples single-step transitions around known trajectories.

My suggestion would be to analyze the implications of the proposed algorithmic components carefully.
1. What is the implication of performing Thompson sampling with a sampling probability dependent on the expected return of some policy?
2. What is the implication of performing Thompson sampling with a sampling probability dependent on the return and reward distribution of some policy (as suggested here).
3. What is the implication of performing Thompson sampling with limited branched rollouts?

Addressing these questions in my opinion would provide a compelling motivation for the proposed algorithm.

---

> ### Author Response · Authors · 2024-11-20
> **Response by Authors**
>
> Thank you very much for taking the time to review our paper. We appreciate your comments and insightful questions. Please find our responses below.
>
> **_Main Weaknesses_**
>
> 1. I am not convinced the reward-uncertainty is significantly more useful than the dynamics uncertainty. As the authors describe, the de-facto standard way of formulating GPs is that output dimensions are considered independent predictions. The uncertainty of each dimension (the variance of the GP) of a GP conditioned on $X$ in a query-point $x$ is then given by $K(x,x) - K(x,X)K^-1(X,X)K(X,x)$ and depends only on the kernel choice and the data distribution. Unless one has specific knowledge about an appropriate reward-kernel, I have no intuition why the reward offers a better (or different) information gain signal than the dynamics.
>
> **Response:** We believe there is some misunderstanding regarding how the GP model is set up. We will try to clarify our model here and have updated the paper to make sure it is better explained. We use a multi-output GP so that our model describes the full covariance structure of the state and the reward as a function of the state and action, i.e. the output dimensions are not conditionally independent given the input. Due to the often high-dimensional state space, we use the linear model of coregionalization to efficiently do computation on this object. As we describe in the paper (lines 270-281), modeling the dependency between predicted state and reward is essential to our approach as this allows us to formulate a distribution that predicts future state based on desired reward. It is correct that we need to make assumptions about the specific relationship between the input (state and action) and the output (next state and reward) for this to be feasible. We use a Matern 5/2 covariance function that encodes a smoothness prior while the coefficients in the linear model of coregionalization are completely free. This means that, if the model is not able to pick up a correlation signal between the output dimensions (reward and state), then its values could be set to zero, implying that the dimensions are independent. In all our experiments, there is a significant correlation between the dimensions, meaning that the reward is informative of the state and vice versa. We believe this is intuitive because the reward is the primary signal that we aim to maximize in RL.
>
> 2. It is unclear to me why the proposed sampling distribution is meaningful. The sampling distribution described in Eq. 8 seems to imply a risk-sensitive sampling strategy without motivating this. For example, one may easily construct MDPs, where optimal trajectories are guaranteed to contain a very small reward. Such trajectories could not be sampled under this strategy, despite being optimal. Mathematically speaking, it is unclear to me what the implications of conditioning on an impossible event is here (i.e., what if $r_t$ can't be greater than $r_{min}$). I can unfortunately not see why one would expect this algorithm to explore reliably.
>
> **Response:** Yes, it is certainly true that these types of problems where optimal trajectories contain very small rewards (i.e., sparse reward settings) are going to be quite challenging for most RL algorithms since exploration is usually done locally. However, we would like to point to the empirical evidence that our approach is doing something sensible and learning effectively even in sparse reward settings, namely the Sparse Reacher variations as well as the new Sparse Maze environments that we have added. We are not making any general theoretical claims about the optimality of our approach and instead show its efficacy on practical, standard benchmarks under difficult learning conditions.
>
> Also, it is important to note that $r_{\text{min}}$ is a percentile value between 0 and 1 (exclusive) so that we draw $r_t$ from the truncated normal distribution $p(r_t \mid s_t, a_t, r_t > r_{\text{min}})$ where $r_t$ is guaranteed to be greater than the value corresponding to the percentile $r_{\text{min}}$. By truncating the distribution, we exclude the region where $r_t$ is less than the value associated with the percentile $r_{\text{min}}$. We are never conditioning on an impossible event where $r_t < r_{\text{min}}$ because there will always be a part of the distribution where $r_t$ is greater than the value corresponding to the percentile $r_{\text{min}}$.

---

> > ### Author Response · Authors · 2024-11-20
> > **Response by Authors - Continued**
> >
> > **_Minor Points_**
> >
> > 1. The dimensions of the GP definition in line 269 don't seem to add up to me. The covariance matrix, as written here, has dimensions $2 \times 2$, while the mean functions are of dimension $|S| + 1$. I moreover think the input dimensions of covariance functions $K$ defined in line 271 should operate on pairs of $S \times A$, not simply $S \times A$.
> >
> > **Response:** Thank you for catching this, yes the input dimensions of covariance functions $K$ should indeed operate on pairs of $S \times A$, not simply $S \times A$. We have changed this in the newly uploaded version of the paper.
> >
> > It is not true that the covariance matrix as given in line 269 has dimensions $2 \times 2$. Given N observations, the full kernel matrix has dimensions $(N \cdot (|S| + 1)) \times (N \cdot (|S| + 1))$ as we model the full covariance structure between the output dimensions. The sub-matrices have the following dimensions: $K_{ss}$ has dimensions $(N \cdot |S|) \times (N \cdot |S|)$, $K_{rr}$ has dimensions $(N \cdot 1) \times (N \cdot 1)$, and $K_{sr}$ has dimensions $(N \cdot |S|) \times (N \cdot 1)$. Additionally, we would like to clarify that $\mu_s$ outputs $|S|$-dimensional vectors and $\mu_r$ outputs scalar values.
> >
> > 2. Expected successor states (as described in 5.2, line 323) don't seem sensible to me. The expectation of a high-dimensional state is not necessarily a meaningful or even existent state. It was moreover not clear to me how the model predictions are conditioned on a certain reward level in practice.
> >
> > **Response:** The advantage of using the GP model is that we have access to the closed form expression of the expectation on line 323. Because our model is a multi-output GP, it describes $s_{t+1},r_t$ as jointly Gaussian, therefore conditioning $s_{t+1}$ on a specific value of the reward can be done in closed form using standard Gaussian conditioning. Since this is all done in closed form, there are no approximations involved.
> >
> > It is absolutely correct that the expectation of a high-dimensional state is not necessarily a meaningful statistic. However, our work demonstrates that our approach is meaningful in practice. Other popular model-based RL works apply similar approaches to predict successor states. For instance, MBPO uses a Negative Log Likelihood loss to learn to predict an independent Gaussian distribution over the reward and dynamics: $p(s_{t+1},r_t \mid s_t, a_t) = \mathcal{N}(\mu(s_t,a_t),\Sigma(s_t,a_t))$ and samples the successor state and reward from this distribution. While handling high-dimensional statistics can be theoretically challenging, such works and ours have shown that this approach can perform effectively in practice.
> >
> > 3. Given that the authors use a GP, I suspect many readers would appreciate a table (or similar) comparing the computational burden of this approach with alternative methods. Furthermore I would be interested to know the effect of subsampling smaller batches (as described in line 850) on the quality of uncertainty estimates with a GP.
> >
> > **Response:** Thank you for the suggestion. We have added Table 2 into Appendix B to compare the average computation speed across each MuJoCo task and algorithm.
> >
> > Regarding the effect of subsampling smaller batches on the quality of uncertainty estimates, this is challenging to assess because of the dependence on the part of the space being explored. Rather than making a specific choice about which points to include, we randomly sample these points. We pick the number 1000 because we found that this balances the computation load with good performance across the board. It is likely that this could be reduced for certain tasks but we did not want this to be a hyperparameter.

---

> ### Author Response · Authors · 2024-11-20
> **Response by Authors - Continued**
>
> **_Questions_**
>
> 1. Why do the authors aim to relieve the stationarity assumption in line 285. Since this GP models the transition dynamics of an MDP, the transition kernel should indeed be stationary. Aiming to address non-stationary MDPs, I believe, would entail significant complications for the entire learning algorithm.
>
> **Response:** We believe there may be some confusion regarding our use of the term "stationary". We refer to non-stationary settings as ones where the dynamics and reward structures vary significantly across different regions of the input space. In GP literature, this is the normal interpretation of the word but we recognize that it can be misleading as this has nothing to do with time. In most GP applications, the mean function is represented in zero or constant form so, if the function is more complex than what the mean can capture, the covariance function has to take responsibility for modeling most of the function's behavior. Instead, our approach uses a neural network as the mean function, which allows the covariance function to focus less on describing the overall behavior of the function. We apologize for the confusion and have removed the word “stationary” from the paper.
>
> 2. In most experiments, the asymptotic performance of the shown algorithm seems to be lower than the base-SAC implementation in all of the environments. Is this due to the cutoff in plotting or is the asymptotic performance indeed lower in most environments? Inserting a dashed line as with SAC into the figure might clear this up.
>
> **Response:** It is true that some training curves do not reach the SAC-best performance level within the maximum number of environment steps. In all experiments apart from the Sparse Reacher, at least one HOT-GP variant (of the three considered $r_{\text{min}}$ schedules) achieves the SAC-best performance after training for a fraction of the environment steps. On the Sparse Reacher task, we believe the HOT-GP performance would continue to increase if we had allowed the algorithm to continue running (i.e., this may be due to the plotting cutoff), however we did not systematically evaluate this due to computational constraints. Note that assessing optimal performance is secondary to the main motivation of our evaluation and we only provide the best-SAC performance to offer an idea of what good performance may be. Our aim in this paper was to evaluate the sample efficiency of HOT-GP relative to existing approaches by comparing algorithms’ performance after a fixed number of environment interactions, as done in H-UCRL (and many other papers).
>
> 3. Given that the authors perform k-branching rollouts, the value estimates bootstrapped at the end of model-generated rollouts should play an important role in quantifying uncertainty. Would omitting this uncertainty skew the ability of the algorithm to explore efficiently? It seems unlikely to me that Thompson sampling maintains its theoretical guarantees if one only samples single-step transitions around known trajectories.
>
> **Response:** This is a really good question that touches on an important avenue worth studying. It is crucial to highlight that our work does not rely on any theoretical guarantees of Thompson sampling, nor do we claim that our algorithm achieves theoretically optimal exploration. Instead, we emphasize the empirical performance of our approach as evidence of its effectiveness.
>
> 4. My suggestion would be to analyze the implications of the proposed algorithmic components carefully.
> What is the implication of performing Thompson sampling with a sampling probability dependent on the expected return of some policy?
> What is the implication of performing Thompson sampling with a sampling probability dependent on the return and reward distribution of some policy (as suggested here).
> What is the implication of performing Thompson sampling with limited branched rollouts?
>
> **Response:** These are extremely interesting questions! We are in complete agreement that modeling the return rather than the reward would allow us to be less myopic in exploration, which would improve sample efficiency even further. However, modeling the full expected return with principled epistemic uncertainty over the dynamics is a very challenging task. This is something that we want to explore in the future, as we believe that some of the concepts around exploration that we have presented in this paper can be useful in that setting.

---

> > ### Comment · Reviewer_QgWe · 2024-11-24
> >
> > Thank you for the detailed and extensive response. I appreciate the time and it has indeed resolved some misunderstandings and points from my initial review. However, I have decided to keep my score at this time, which is mainly due to (a combination of) two reasons:
> >
> > 1. As pointed out in my review, I think this paper would benefit greatly from a stronger theoretical justification of the proposed methods and ideas. I generally find the avenue of return-conditioned rollouts with learned models an interesting and important one.
> >
> > 2. For a more empirically motivated paper, I think the experimental section would need to be exceptionally strong. In its current form, however, I find the empirical benefit not quite convincing enough (both in the performance on the shown experiments and in terms of the variety of experiemnts) for the high requirements of this venue.

---

> ### Author Response · Authors · 2024-11-27
> **Response by Authors**
>
> Thank you for your response. We are glad to hear that our explanations have helped resolve some misunderstandings and points from your original review. We appreciate the additional feedback and aim to address these points individually.
>
> 1. As pointed out in my review, I think this paper would benefit greatly from a stronger theoretical justification of the proposed methods and ideas. I generally find the avenue of return-conditioned rollouts with learned models an interesting and important one.
>
> **Response:** Thank you for the suggestion. We would like to highlight some theories that our work currently builds upon.
>
> For a high-level theoretical justification of HOT-GP, we refer to the thresholded Thompson sampling regret bounds presented in [1]. In a combinatorial semi-bandit setting, the authors show that regret is no longer dependent on $T$, the number of timesteps, when the rare arms are never played. This is similar to how we perform optimistic Thompson sampling by drawing rewards from a truncated normal distribution, where the region where $r_t$ is less than the value associated with the percentile $r_{\text{min}}$ is excluded. For Thompson sampling regret bounds on reinforcement learning problems, we refer to the theoretical analysis in [2].
>
> Regarding the performance implication of using limited k-branched rollouts, we refer to Theorem 4.3 in [3]. This theorem states that, as long as the returns of the policy under the estimated model are lower-bounded by $2r_{\text{max}} \left[\frac{\gamma^{k+1}\epsilon_{\pi}}{(1-\gamma)^2} + \frac{\gamma^{k}\epsilon_{\pi}}{(1-\gamma)} + \frac{k}{1-\gamma}(\epsilon_{m’}) \right]$, performance improvement on the true MDP is guaranteed. Here $\epsilon_{\pi}$ and $\epsilon_{m’}$ denotes the distribution shift of the policy and generalization error between iterations respectively. Notably, since the argmin with respect to $k$ of $ \left[\frac{\gamma^{k+1}\epsilon_{\pi}}{(1-\gamma)^2} + \frac{\gamma^{k}\epsilon_{\pi}}{(1-\gamma)} + \frac{k}{1-\gamma}(\epsilon_{m’}) \right] > 0$, this theoretically motivates the use of k-branched rollouts. Going back to one of the questions raised in the original review, the implication of doing this with Thompson sampling is that Thompson sampling may not be able to explore the full range of outcomes, yet we get to benefit from the improvement in performance. We empirically study this tradeoff with an ablation study in Figure 3, showing that performing Thompson sampling over $k=5$ branched rollouts and full rollouts with $k=T$ (task horizon) significantly harms learning, likely due to accumulated model bias that prevents performance improvement in the true MDP.
>
> Regarding return-conditioned rollouts with learned models, we agree that this is an interesting and important avenue. In our work, we treat the single step reward values as a proxy for the return and condition transitions on optimistic and feasible rewards. A natural next step would be to extend the optimistic horizon to match the task horizon and condition entire rollouts on optimistic total returns. In theory, this could enable the discovery of optimal trajectories, where intermediate states yield small rewards and later states provide larger rewards (as pointed out in your original review). Such exploration can be driven by epistemic uncertainty over performance as encoded over multi-step imagined returns or value functions. However, these approaches remain quite challenging due to the accumulation of model biases over multi-step imagined rollouts [3] and the rigidity of overestimated value functions [4, 5]. For instance, the recent method presented in [6] aims to be optimistic over the values yet does not reach SAC-best performance on some standard benchmark tasks. While this line of work is very promising, further research is needed to develop algorithms that can effectively explore using optimistic return-conditioned rollouts. By establishing the importance of joint uncertainty modeling for optimistic exploration, we believe that our work meaningfully contributes to advancing this area and we are eager to extend our work in this direction in the future. However, due to the many challenges involved, directly addressing optimistic return-conditioned rollouts is out of scope for this paper.

---

> ### Author Response · Authors · 2024-11-27
> **Response by Authors - Continued**
>
> 2. For a more empirically motivated paper, I think the experimental section would need to be exceptionally strong. In its current form, however, I find the empirical benefit not quite convincing enough (both in the performance on the shown experiments and in terms of the variety of experiments) for the high requirements of this venue.
>
> **Response:** Regarding the experimental section, we would like to highlight that this has been improved in the newest version of the paper that was updated after your most recent comment. These updates directly respond to both criteria you mentioned: the variety of experiments and the performance on shown experiments.
>
> First, following Reviewer LvwR’s suggestion, we have evaluated HOT-GP on two Sparse Point Mazes of varying difficulty from the D4RL suite (Figure 2). In the simpler U Maze environment, HOT-GP matches the sample efficiency and performance of alternative methods. In the more challenging Medium [sized] Maze environment, HOT-GP with higher levels of optimism ($r_{\text{min}}$ from 0.1 to 0.7) outperforms both HOT-GP with lower levels of optimism and all comparison methods. These results validate the findings from our original experiments and demonstrate HOT-GP’s adaptability to new, complex tasks requiring long-horizon exploration, all without requiring tuning $r_{\text{min}}$ or other parameters. Notably, we observe consistency in these results across diverse test suites. If there are additional specific experiments that you think would highlight particular characteristics of our model or approach, we would be happy to do such evaluations for the final version of the paper.
>
> Secondly, to further address Question 2 in your original review, we reran our experiments on Sparse Reacher with more environment interactions to clarify the asymptotic performance (Figure 6). These results show that HOT-GP converges to SAC-best performance after just 1/10th of the environment steps. Across other tasks, our original results already show that at least one HOT-GP variant achieves performance comparable to the best model-free asymptotic results.
>
> We hope that our response has sufficiently addressed your concerns, but if you have any further comments or questions please do not hesitate to ask. We believe that these additions strengthen the experimental section of the paper as they showcase both optimal policy convergence and robust performance across a diverse range of experiments. We would be grateful if you would consider increasing your score in light of these responses. Thank you!
>
> [1] Bubeck and Sellke. First-order bayesian regret analysis of thompson sampling. Algorithmic Learning Theory, 2020.\
> [2] Moradipari et al. Improved bayesian regret bounds for thompson sampling in reinforcement learning. NeurIPS, 2023.\
> [3] Janner et al. When to trust your model: model-based policy optimization. NeurIPS, 2019.\
> [4] Sims et al. The edge-of-reach problem in offline model-based reinforcement learning. NeurIPS, 2024.\
> [5] Fujimoto et al. Addressing function approximation error in actor-critic methods. ICML, 2018.\
> [6] Luis et al. Model-based epistemic variance of values for risk-aware policy optimization. arXiv, 2024.

---

> > ### Author Response · Authors · 2024-12-02
> > **Follow Up by Authors**
> >
> > As the end of the discussion period is approaching, we wanted to follow up on our last response. We believe we have addressed your feedback by (1) referencing some theoretical justifications for our approach and (2) strengthening the experiments, which now include a comparison of the asymptotic performance on the Sparse Reacher task and additional evaluations on two Sparse Point Maze problems. If the empirical benefit was “not quite convincing enough” before, we hope it is more convincing with these improvements. If so, we would greatly appreciate it if you would consider updating your score to reflect that. Thank you!

---

### Official Review · Reviewer_ab8V · 2024-11-04

**Soundness:** 2
**Presentation:** 3
**Contribution:** 2
**Rating:** 8
**Confidence:** 3

**Summary:**

This paper proposed Hallucination-based Optimistic Thompson Sampling with Gaussian Processes (HOT-GP),  a method for principled optimistic exploration. It uses a GP with Thompson Sampling as the acquisition function to maintain a joint belief over state and reward distributions, enabling simulating optimistic transitions wrt the estimated reward. Experiments on MuJoCo and VMAS continuous control tasks demonstrate that the proposed approach matches or outperforms baselines, showing substantial improvements in challenging exploration settings. The paper also investigates factors influencing optimism through ablation studies, highlighting the importance of joint uncertainty modelling in guiding effective exploration.

**Strengths:**

- principled approach for exploration in continuous state-action space
- the first method to predict joint uncertainty over next state and reward

**Weaknesses:**

- The description for Fig.2 in Section 6.2, references several methods that are not shown in the plots. For instance,
    - SAC and PPO as described in Line 430, “we use … SAC, and the on-policy PPO”
    - HOT-GP with k-branching (k=5) as described in Line 470.
- Learning curves seem to be from the training episodes. It would be better to report the performance in evaluation episodes instead, as done in SAC.
- The method introduces a crucial hyperparameter reward threshold r_{min}, which controls optimism during exploration. While the results do not seem to be overly sensitive to it, its effect in other tasks remains unclear. Although future work is mentioned to address this, its current implementation could limit its applicability.

**Questions:**

Could you provide any results on computational cost or computation speed for the proposed method?

---

> ### Author Response · Authors · 2024-11-20
> **Response by Authors**
>
> Thank you very much for taking the time to review our paper. We appreciate your comments and thoughtful feedback. Please find our responses below.
>
> **_Weaknesses_**
>
> 1. The description for Fig.2 in Section 6.2, references several methods that are not shown in the plots. For instance, SAC and PPO as described in Line 430, “we use … SAC, and the on-policy PPO” and HOT-GP with k-branching (k=5) as described in Line 470.
>
> **Response:** Thank you for pointing this out and we apologize for the confusion. We have uploaded a new paper pdf in which Figure 3 (the old Figure 2) has been updated with the correct corresponding training curves and algorithms.
>
> 2. Learning curves seem to be from the training episodes. It would be better to report the performance in evaluation episodes instead, as done in SAC.
>
> **Response:** Our learning curves actually show the total reward averaged across evaluation episodes (not training episodes). We agree this was initially unclear and have clarified this metric in the new version of the paper.
>
> 3. The method introduces a crucial hyperparameter reward threshold $r_{min}$, which controls optimism during exploration. While the results do not seem to be overly sensitive to it, its effect in other tasks remains unclear. Although future work is mentioned to address this, its current implementation could limit its applicability.
>
> **Response:** Indeed, we find that $r_{\text{min}}$ is not very challenging to set in practice, although setting this optimally is an open question for future research. Currently, we use a linear scheduler to increase $r_{\text{min}}$ during training. This effectively increases the level of optimism during training, as too much early/naive optimism could be detrimental to the learning process. The most principled approach to setting $r_{\text{min}}$ would be to set a distribution over $r_{\text{min}}$ and integrate it out, however this cannot be done in closed form and hence is computationally expensive. We plan to look into approximations that enable learning $r_{\text{min}}$ adaptively in future work. Note that it is common in the literature to introduce a hyperparameter to balance the trade-off between exploitation and exploration, as seen in H-UCRL with the $\beta$ hyperparameter. However, a key distinction with our approach is that the results are not very sensitive to the hyperparameter's value. In contrast, our experiments with H-UCRL revealed a strong dependence between the performance and the $\beta$ value.
>
> Overall, we completely agree that better understanding the implications of this parameter is important. In this work, we want to convey that a non-zero $r_{\text{min}}$ value improves performance and, crucially, that this effect is robust across different tasks. Our results on the MuJoCo and VMAS tasks already highlight this robustness, and we are currently running additional experiments on two Sparse Point Mazes. Our results on these tasks so far also indicate consistent performance across various $r_{\text{min}}$ values, providing further evidence that our approach is robust to the specific value.
>
> **_Questions_**
>
> 1. Could you provide any results on computational cost or computation speed for the proposed method?
>
> **Response:** Certainly, we have added Table 2 into Appendix B comparing the average computation speed across each MuJoCo task and algorithm.

---

> > ### Comment · Reviewer_ab8V · 2024-11-26
> >
> > I have carefully reviewed the authors' response and the comments from other reviewers. The related work highlighted by Reviewer LvwR looks interesting and indicates that this work may not be the first to predict joint uncertainty over the next state and reward, contrary to my initial impression.
> > Despite this, I appreciate the thorough clarifications and revision made to the paper.
> > I find that the manuscript has been improved during the rebuttal phase, and hence I have increased my score.

---

> ### Author Response · Authors · 2024-11-27
> **Response by Authors**
>
> Thank you very much for your thoughtful follow-up review and increasing your score! We are pleased to hear that our revisions addressed your concerns and that you found the manuscript stronger as a result.
>
> Regarding the papers highlighted by Reviewer LvwR, we have clarified their relationship to our work and also restated the contributions and novelty of our submission in our response. We are happy to engage further about this if you have any questions or comments. Additionally, please let us know if there is any other information we can provide to help strengthen the confidence of your recommendation. Thank you!

---

### Author Response · Authors · 2024-11-25
**Updated Paper Submission**

We would like to thank the reviewers again for their time and valuable comments. A revised version of the manuscript has been uploaded, incorporating the following main changes marked in maroon color:

- Conducted additional experiments on two Sparse Point Mazes from the D4RL suite (Figure 2)
- Extended the Sparse Reacher experiments, demonstrating that HOT-GP’s asymptotic performance matches that of SAC (Figure 6)
- Added an extra experiment on Sparse Reacher with the intermediate action penalty of 0.3 (Figure 7)
- Increased the number of seeds for MuJoCo experiments to 10 per algorithm (Figure 1)
- Updated results on the Coverage task with the correct corresponding algorithms’ performance (Figure 3)
- Provided details on using neural networks as GP mean functions (Appendix B)
- Inserted Table 2 to specify the average runtime for each MuJoCo task and algorithm (Appendix B)

We believe that these changes have substantially enhanced the quality of the paper and invite the reviewers to reconsider our submission, potentially adjusting their scores. Notably, our evaluations across three test suites demonstrate the benefits of our novel approach, namely modeling and leveraging optimism with joint uncertainty. These advantages are particularly pronounced in challenging learning settings involving sparse rewards, difficult exploration, and action penalties. If there are any remaining concerns, please let us know and we will address them!

---

### Meta-Review · Area_Chair_tuu3 · 2024-12-22

**Metareview:**

The paper proposes HOT-GP, a model-based Thompson sampling-based RL method that takes into account the *joint* uncertainty in dynamics and reward, as modeled by a joint Gaussian Process, when making exploration decisions. The empirical evaluation shows it to be generally more sample-efficient than several baselines, including vanilla SAC, on MuJoCo and maze tasks.

Strengths:
- Novelty of the sampling distribution for exploration
- Good practical implementation
- Clarity

Weaknesses:
- Not entirely clear why it works better than other exploration types, since the uncertainty is estimated over instantaneous reward, not cumulative return
- The empirical evaluation isn't very strong, done on simple benchmarks, although it has definitely improved after the discussion with the reviewers

The metareviewer recommends this work for acceptance due to the novelty of the method, the empirical results showing its promise, and further analysis/discussion it can spark in the community. The metareviewer partly agrees with reviewer *QgWe* that it's not obvious why exploring based on uncertainty in both reward and dynamics gives better results than based on uncertainty in the dynamics alone. One possible explanation is that this simply allows RL to have a more complete picture of uncertainty and hence resolve the uncertainty about the model more quickly, more even if the uncertainty is over instantaneous reward rather than cumulative return. In any case, the metareviewer doesn't view the lack of complete clarity in this aspect as a blocker for publication, but rather as something that will give the research community food for further thought.

**Additional Comments On Reviewer Discussion:**

The discussion revolved mostly around the empirical evaluation and explanations of HOT-GP's superior results. As a result, the paper has improved in both aspects, even if the theoretical explanations aren't fully conclusive. As mentioned above, the metareviewer doesn't view a fully conclusive theoretical explanation of HOT-GP's behavior as a condition for publication.

Reviewer *LvwR* also views HOT-GP's novelty compared to H-UCRL as limited. The metareviewer doesn't disagree entirely but thinks that the analysis of such a sensibly-looking proposal distribution as the one based on joint uncertainty in dynamics and reward is sufficiently valuable for publication.

---

### Decision · Program_Chairs · 2025-01-22

Accept (Poster)